# CFDLLMBench: A Benchmark Suite for Evaluating Large Language Models in Computational Fluid Dynamics

## Abstract

Large Language Models (LLMs) have demonstrated strong performance across general NLP tasks, but their utility in automating numerical experiments of complex physical system—a critical and labor-intensive component—remains underexplored. As the major workhorse of computational science over the past decades, Computational Fluid Dynamics (CFD) offers a uniquely challenging testbed for evaluating the scientific capabilities of LLMs. We introduce *CFDLLMBench*, a benchmark suite comprising three complementary components—*CFDQuery*, *CFDCodeBench*, and *FoamBench*—designed to holistically evaluate LLM performance across three key competencies: graduate-level CFD knowledge, numerical and physical reasoning of CFD, and context-dependent implementation of CFD workflows. Grounded in real-world CFD practices, our benchmark combines a detailed task taxonomy with a rigorous evaluation framework to deliver reproducible results and quantify LLM performance across code executability, solution accuracy, and numerical convergence behavior. *CFDLLMBench* establishes a solid foundation for the development and evaluation of LLM-driven automation of numerical experiments for complex physical systems.

## 1 Introduction

Recent advances in large language models (LLMs) have shown remarkable performance across general natural language processing tasks (Grattafiori et al., 2024; Achiam et al., 2023). However, their potential as scientific assistants—specifically, their ability to automate numerical simulation workflows—remains largely underexplored (Bran et al., 2023; Kumar et al., 2023).

Computational Fluid Dynamics (CFD) is critical in domains such as urban physics (Blocken et al., 2011; Blocken, 2015), aerospace (Slotnick et al., 2014), climate (Shah et al., 2023), aerial (Shi et al., 2019), underwater robotics (Lee et al., 2023a), and has labor-intensive workflows for computationally expensive numerical simulations of fluid dynamics. CFD workflows involve multiple steps, such as mesh generation, setup of boundary and initial conditions, and solver configuration. Such scientific workflows require an understanding of highly specialized knowledge (Wang et al., 2023a), numerical and physical reasoning (Tian et al., 2024), and have context-dependent implementations involving domain-specific tool calling (Jacobs & Pollice, 2025).

In this paper, we introduce *CFDLLMBench* (Figure 1), the first LLM benchmark for CFD composed of curated datasets designed to holistically evaluate LLMs' performance across three key competencies:

**Graduate-level CFD knowledge:** Understanding of fluid mechanics and concepts of numerical analysis relevant to CFD.

**Numerical and physical reasoning:** Applying advanced math and physics knowledge to solve difficult problems. For example, selecting a suitable numerical method that solves the governing equation, with the appropriate boundary conditions and initial conditions.

**Context-dependent implementation of CFD workflows:** Selecting and configuring CFD preprocessing and numerical solver settings according to the physical context.

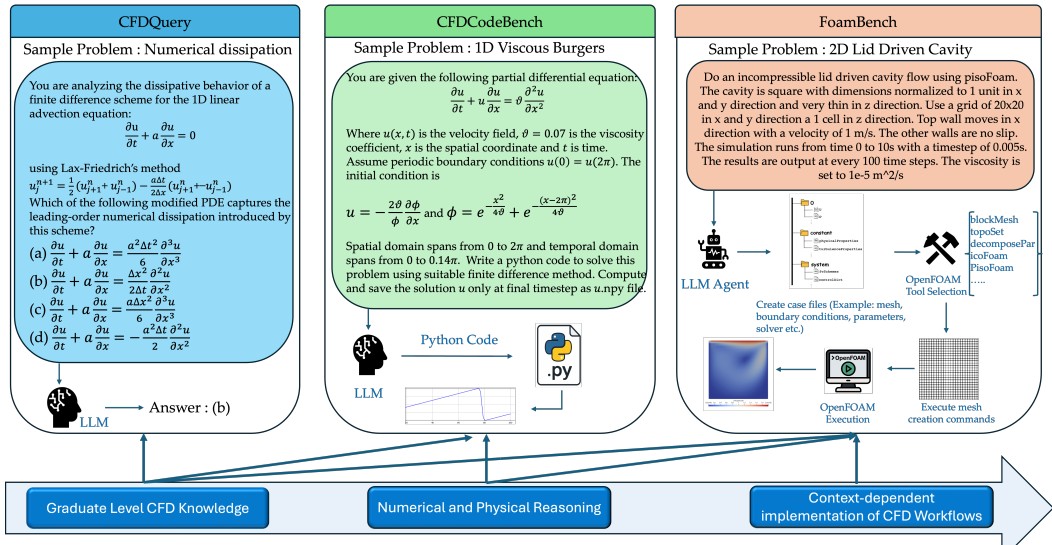

**Figure 1: Overview of CFDLLMBench**: As the first ever LLM benchmark designed to holistically evaluate LLM's capabilities for CFD, it consists of three different tasks and datasets. (1) *CFDQuery*: Graduate-level CFD QA. (2) *CFDCodeBench*: Coding questions about solving common linear/nonlinear PDEs encountered in CFD. (3) *FoamBench*: Configuring OpenFOAM case files for simulating realistic engineering scenarios such as incompressible flow over obstacles, supersonic flow with shockwaves, Rayleigh-Benard convection, etc.

The *CFDLLMBench* benchmark suite evaluates these competencies using three benchmark tasks: **1) CFDQuery:** 90 multiple-choice questions curated from graduate-level CFD lecture notes that assess LLM's ability in the conceptual understanding of CFD knowledge. **2) CFDCodeBench:** 24 CFD programming tasks designed to assess an LLM's ability to generate correct simulation code from descriptions of physical problems. **3) FoamBench:** 110 basic and 16 advanced numerical simulation tasks, drawn from practical engineering problems, designed to assess the LLM's ability to implement OpenFOAM (Weller et al., 1998) workflows. OpenFOAM projects typically have 6-7 configuration files, totaling ∼300-600 lines of code per case.

Although strong performance on *CFDQuery* indicates excellent recall of relevant CFD knowledge, success in solving *CFDCodeBench* and *FoamBench* would suggest that LLM possesses reasoning and workflow implementation capabilities near the proficiency of a competent CFD assistant. To support a holistic evaluation of these diverse benchmark tasks, we equip each benchmark task with one or more tailored metrics, which are developed in collaboration with CFD experts.

We use *CFDLLMBench* to evaluate both state-of-the-art proprietary and open-source LLMs. Despite relatively strong performance on *CFDQuery*, the results highlight the challenge of the latter two tasks (see Figure 2): the best performing model achieves only **14%** on *CFDCodeBench* and **34%** on *FoamBench*. In the more complex *FoamBench Advanced* split, generally, performance is poor, e.g., Gemini 2.5 Flash drops to **0%**. In *FoamBench*, all models show major improvement when deployed in a multi-agent framework, as opposed to zero-shot prompting (near 0 performance).

The remainder of the paper is organized as follows. Section 2 describes related work. Section 3 presents our holistic CFD benchmark. Section 4 summarizes our experimental setup and results, which are discussed in Section 5. Section 6 has limitations and Section 7 concludes the paper.

## 2 RELATED WORK

**LLMs for science & engineering** LLMs are becoming increasingly proficient at knowledge-intensive tasks in general science (Taylor et al., 2022; Beltagy et al., 2019; Luo et al., 2022; Singhal et al., 2025) and engineering (Jadhav & Farimani, 2024), aided by dedicated pretraining on scientific corpora. The development of language agents with tool-use (Qin et al., 2023; Boiko et al., 2023; Bran

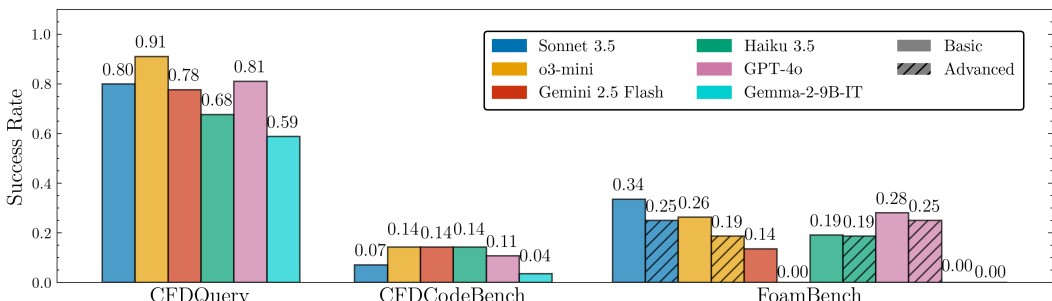

**Figure 2:** Success Rate comparison of different models across the three tasks. Success Rate is the fraction of cases in the benchmark that produce physically accurate results (higher is better). The detailed definition of Success Rate for each benchmark task can be found in section 3.3. The results for *FoamBench* are produced using the *Foam-Agent* framework with RAG, Reviewer, and Sonnet 3.5. There is a steep drop in performance from graduate-level knowledge (*CFDQuery*) to practical simulation workflow automation *FoamBench*.

et al., 2023; Narayanan et al., 2024) further enhances LLMs' capabilities, enabling them to integrate with complex scientific and engineering software (Cherian et al., 2024).

Recent work explores the use of LLMs to generate input files in domain-specific languages for quantum chemistry (Jacobs & Pollice, 2025) and building energy (Jiang et al., 2024) simulators, tasks which demand substantial time from a researcher to master. LLMs are also accelerating workflow automation in computational physics. Studies such as (Kashefi & Mukerji, 2023) and (Jiang et al., 2025) demonstrate the usage of LLMs to numerically solve simple PDE problems. MyCrunchGPT (Kumar et al., 2023) demonstrates the use of automated scientific machine learning workflows to optimize airfoils in aerodynamics. MetaOpenFOAM (Chen et al., 2024a), OpenFOAMGPT (Pandey et al., 2025), and Foam-Agent (Yue et al., 2025) exemplify this trend by automatically configuring and conducting complex CFD simulations based on human requests. These examples highlight the critical need for effective workflow automation benchmarking.

**LLM benchmarks for science & engineering**   Recent interest in the use of LLMs in science and engineering has led to benchmarks measuring specific advanced LLM capabilities such as graduate-level scientific problem solving (Rein et al., 2024; Wang et al., 2023b; Glazer et al., 2024; Zhang et al., 2025) and long-context reasoning (Lee et al., 2023b; Cui et al., 2025). Our benchmark aims at practicality, providing a holistic evaluation that includes a real-world numerical simulation workflow automation task. Other related workflow benchmarks focus on paper reproduction (Starace et al., 2025; Bogin et al., 2024; Siegel et al., 2024) or data analysis workflows (Chen et al., 2024b; Majumder et al., 2024; Mitchener et al., 2025). Paper reproduction, data analysis, and simulation automation (ours) are all critical workflows in the scientific discovery life cycle. Differently, our benchmark uniquely evaluates numerical and physical reasoning, an underexplored capability in LLMs. Thus, these benchmarks assess distinct yet complementary capabilities for scientific workflow automation. The most closely related benchmark is FEABench (Li et al., 2025), which evaluates the ability of LLMs as agents for solving PDEs using COMSOL, a commercial finite element analysis software that requires a license of several thousand dollars per year. In contrast, our work is a comprehensive benchmark that consists of domain-specific knowledge, reasoning, and OpenFOAM (Jasak et al., 2007)workflow automation, one of the most widely used open-source numerical simulation software.

**LLM benchmarks for code generation**   Code generation benchmarks such as MBPP (Austin et al., 2021), HumanEval (Chen et al., 2021), DS-1000 (Lai et al., 2023), and SWE-Bench (Jimenez et al., 2023) evaluate general coding yet lack the complexity of scientific and engineering tasks. These require understanding advanced concepts and implementing sophisticated algorithms that involve specialized libraries. SciCode (Tian et al., 2024) is a related scientific coding benchmark, but their CFD examples-1D heat transfer and 1D Burgers equation-are far from enough to represent the algorithmic, physical, and geometrical complexity in CFD. There is a clear need for a comprehensive code generation benchmark that meets the scientific standards for CFD.

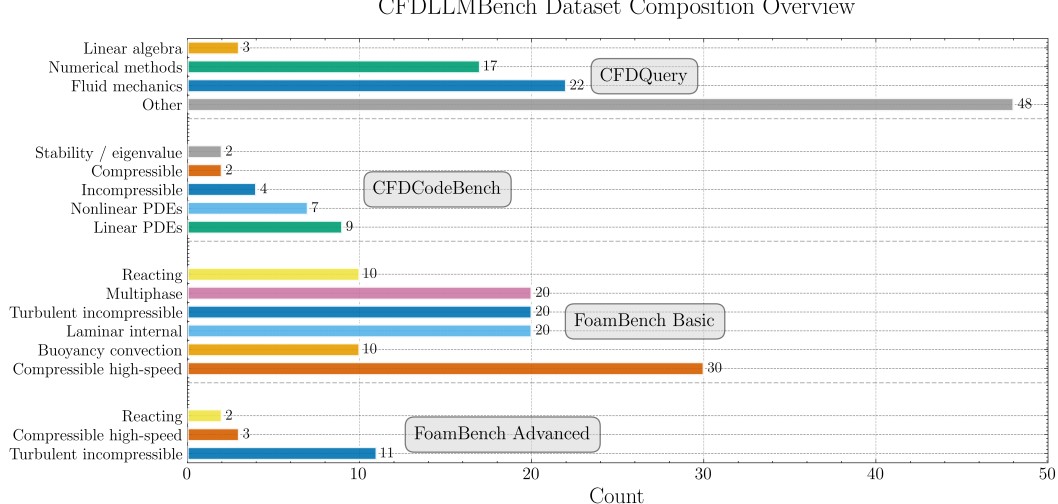

**Figure 3:** Distribution of cases within CFDLLMBench. The problems span simple theortical questions to 1D/2D PDE problems to partial simulations tasks in different flow regimes.

## 3 CFDLLMBENCH: A BENCHMARK SUITE FOR EVALUATING LLMS IN CFD

We present *CFDLLMBench*, which holistically assesses three capabilities of LLMs necessary to perform CFD-related tasks (Figure 1). We begin with *CFDQuery* which evaluates graduate-level conceptual understanding, after which the benchmark progresses to the application of this knowledge through *CFDCodeBench*, where LLMs must use numerical and physical reasoning over a description of a physical problem to correctly generate CFD code in Python. Finally, the most practical and challenging benchmark task is *FoamBench*, where LLMs write input files for a CFD software suite that must correctly pre-process, configure, and execute simulations given physical context expressed in natural language.

**OpenFOAM**  OpenFOAM (Weller et al., 1998) is an open-source, license-free CFD software suite (a collection of software for fluid-flow simulation that covers meshing, solving, and post-processing) widely used in academia and industry. OpenFOAM projects have a precise file organization and various configuration and source files arranged in a strict folder hierarchy. OpenFOAM's accessibility, extensibility, and rich community resources make it an attractive platform for an LLM benchmark. However, automating OpenFOAM workflows poses significant challenges for language models and agents. Writing code for OpenFOAM requires long-context understanding to track simulation parameters across multiple files, domain-specific tool usage, and accurate implementation of complex physical models. The third benchmark task in our suite, *FoamBench*, uses OpenFOAM as the underlying CFD software suite.

### 3.1 DATASETS OVERVIEW

To contextualize the diversity of tasks in CFDLLMBench, we provide an illustration of the categories of problems, ranging from theoretical reasoning, to code generation for PDE solution, to full OpenFOAM simulation workflows in fig. 3. Below, we summarize the design and scope of each dataset.

**CFDQuery**  This dataset consists of 90 multiple-choice questions pertaining to CFD curated by three domain experts. These questions probe core concepts in fluid mechanics, linear algebra, and numerical methods, with source materials adapted from both web-scraped content and CFD lecture notes. The solution to these problems require the LLMs to have deep knowledge about topics in CFD like linear algebra, numerical methods and fluid dynamics.

**CFDCodeBench**   This dataset consists of 24 CFD problems that require LLMs to generate Python code for their numerical solution. Each problem is described in natural language and specifies the governing Partial Differential Equation (PDE), boundary and initial conditions, the spatial and temporal domain, and the target variable(s) to be computed and saved. The dataset includes both 1D and 2D problems, spanning linear and nonlinear PDEs, representative of those encountered in the CFD domain. Reference solutions are provided either as closed-form analytical expressions or as expert-authored Python implementations. Further details can be found in Appendix A.3.2. Our 24 coding problems span fluid mechanics, thermal transport, and turbulence, include both 1D and 2D simulation scenarios, extending beyond prior work in terms of complexity, which only evaluates the 1D heat transfer and 1D Burgers' equation (Tian et al., 2024), in both scope and complexity. Solving these problems requires not only reasoning about the physics but also integrating numerical methods, discretization schemes, and data handling into coherent, executable Python scripts containing 70 lines of code on average per problem.

**FoamBench**   This task requires LLMs to generate all input files for an OpenFOAM simulation using the proper project folder structure and for the simulation to execute correctly, producing a physically accurate result with respect to a reference project. It consists of 126 OpenFOAM cases spread over more than 15 distinct geometric and physics scenarios. This dataset is further divided into two. *(1) FoamBench Basic*: This consists of 110 OpenFoam cases obtained from 11 tutorial cases (Weller et al., 1998). We create variations within them by altering the boundary conditions and the parametric values on a case-specific basis (more details can be found in Appendix A.3.3). *(2) FoamBench Advanced*: This consists of 16 challenging OpenFOAM cases, which are not similar to the tutorials and are hand-crafted by CFD experts. Unlike *Basic*, the *Advanced* split tasks LLMs with choosing a proper turbulence model, creating a new geometry, and creating an appropriate mesh, based on the natural language input, without potentially relying on a tutorial project for guidance. For example, in the *Advanced* flow over double square case, the prompt specifies two square obstacles with details of their location and size. The LLM must correctly understand this prompt, then use appropriate one or more meshing tools from the OpenFOAM suite (e.g. blockMesh) to generate a valid computational mesh. Such cases bring us closer to real-world scenarios, where engineers analyze flow over complex geometries based on design specifications. Further details of the cases are provided in Appendix A.3.3.

For each case in *FoamBench*, the prompt (Appendix A.3.3) is designed to be concise and sufficient. The prompt contains (1) a clear description of the problem (e.g., flow over a cylinder), physical scenario (compressible or incompressible), geometry including computational domain and obstacle locations (with retrieval mechanisms handling complex geometries) and specifies the exact Open-FOAM solver for consistency; (2) the boundary conditions, relevant parameters (viscosity, Prandtl number), turbulence models (e.g., $k - \epsilon$, SA, LES), and specifies the timestep and solution-saving intervals for comparison against reference solutions.

## 3.2   DATASET CREATION

In this section, we describe the dataset curation process. Due to the complex and technical nature of our benchmark, we relied on human experts at several stages during the creation of *CFDLLMBench*, involving them in both curation of data from existing sources, as well as authoring new content for the benchmark. A complete description of our process is presented in Appendix A.3.

**Expert contributors**   For all three datasets, human experts curated or authored the initial set of problems. Our team included six domain experts with advanced degrees and professional experience in the field of CFD. Despite being experts in CFD, they were still provided an orientation ahead of the curation process. For *CFDQuery*, the human experts created the multiple choice problems, and for *CFDCodeBench*, the human experts authored descriptions for the advanced problems by reviewing the source code. For *FoamBench*, the experts curated the dataset by varying parameters and boundary conditions for the tutorial problems, designing novel geometries for the non-tutorial cases, and authoring corresponding prompts based on the case files to guide LLMs in generating valid simulation setups. While the nature of the human work did not warrant an IRB review, we nevertheless followed all ethical norms and standards of the host academic institute when performing the human tasks for this dataset.

**Data sources**  For this benchmark, we ensured that we only used highly vetted data sources. The *CFDQuery* dataset was created exclusively for this benchmark, but the reference sources include university-level CFD lecture notes and vetted online sources. The problems in *CFDCodeBench* were curated from publicly available GitHub repositories and established numerical solver packages, including *CFD Python: the 12 Steps to Navier-Stokes Equations* repository (Barba & Forsyth, 2018) and *ENGR 491 - Computational Fluid Dynamics*, while more challenging scenarios were curated from the Dedalus Project (Burns et al., 2020). For *FoamBench*, we curated the dataset based on the 11 OpenFOAM tutorials (Weller et al., 1998).

**Quality assessment**  Since the solutions to our problems include objective, scientific answers, we did not perform traditional measures of human agreement. Rather, we went through an iterative process of review and revision of human work by independent experts to ensure the quality of the work. This review included both human-curated and human-authored portions of the benchmark.

## 3.3  EVALUATION METRICS

Here we define expert-informed metrics used to assess performance on *CFDLLMBench*.

**CFDQuery**  We evaluate multiple choice accuracy using a single standard accuracy metric, *Success Rate*, defined as ratio of the number of correctly answered questions to the total number of questions.

**CFDCodeBench**  We evaluate an LLM's ability to generate executable and physically accurate python code for the numerical solution of a given CFD problem using four metrics. The holistic metric we use has three components: code executability, relative numerical error, and numerical convergence. We aggregate these three into a single score, which we call the *Success Rate*. **1) Executability** ($M_{\text{exec}}$): This is a binary metric which takes on a value of 1 if the LLM generated python code executes successfully and 0 if it is a failure. This metric is akin to the common pass@1 metric (Chen et al., 2021). **2) Relative Error** ($M_{\text{NMSE}}$): We compare the LLM generated solution to the reference solution at the *final time of the prescribed simulation interval*. A normalized mean squared error percentage is calculated and a score is assigned based on the value of the NMSE percentage given by

$$\text{NMSE\%} = \frac{\sum_{i=1}^{N}(y_i - \hat{y}_i)^2}{\sum_{i=1}^{N} y_i^2} \times 100, \quad M_{\text{NMSE}} = \begin{cases} 1, & \text{NMSE} \leq 10\%, \\ 0.5, & 10\% < \text{NMSE} \leq 30\%, \\ 0, & \text{NMSE} > 30\%. \end{cases} \quad (1)$$

An $M_{\text{NMSE}}$ of 0 means the solution is not physically accurate, a score of 0.5 is considered partial success and a score of 1 means the solution is acceptably accurate. The NMSE thresholds of 10% and 30% are not chosen arbitrarily, instead they are grounded in engineering practice and further supported by an empirical sensitivity analysis with details provided in Appendix A.2. **3) Numerical convergence** ($M_{\text{conv}}$): To evaluate the numerical convergence of the solution generated by the LLM, we refine both the spatial and temporal discretization and assess the corresponding change in relative error. If the error decreases with mesh and time-step refinement, the solution is deemed convergent and awarded a score of 1; otherwise, it receives a score of 0. Unlike conventional LLM code generation benchmarks, we cannot rely on code similarity with respect to a reference solution, as numerical simulation code can vary significantly in implementation while yielding identical or equivalent solutions. **4) Success Rate**: We also define a stringent criterion to assess successful runs by looking at the fraction of problems where *all three* metrics achieve a score of 1. Specifically, defined for each problem $i$:

$$M_{\text{success}}^{(i)} = \begin{cases} 1, & M_{\text{exec}}^{(i)} = 1 \ \wedge \ M_{\text{NMSE}}^{(i)} = 1 \ \wedge \ M_{\text{conv}}^{(i)} = 1, \\ 0, & \text{otherwise}, \end{cases} \quad \text{Success Rate} = \frac{1}{K} \sum_{i=1}^{K} M_{\text{success}}^{(i)},$$

$$(2)$$

where $K$ is the total number of problems. This provides us with a stringent measure of the percentage of problems within the benchmark where the model was able to produce an executable, physically accurate, and convergent solution.

Our evaluation focuses on results, not specific solution methods. If an LLM produces code using a different but valid numerical algorithm (e.g., finite difference vs. finite volume), it is still accepted as

correct provided that the numerical error remains below tolerance and solution converges. Hence, the benchmark does not penalize algorithmic diversity, only the correctness and stability of results.

**FoamBench**  This task requires an LLM to create the required OpenFOAM input files, save them in appropriate directories, and call different tools within OpenFOAM to run a physically accurate simulation, all based on a natural language prompt. Prior work (Chen et al., 2024a) focuses only on the ability of LLMs to generate files that produces a successful execution of OpenFOAM. Though executability is important, it does not capture the physical accuracy of the generated solution and thus fails to provide insights into whether the solution satisfies the user requirements. Text similarity metrics are widely used in comparing LLM-generated text to human text. For code generation, this is a useful metric for giving us an idea of how complete the files generated by LLMs are in comparison to the reference files, but again fails to provide the complete picture.

To tackle these challenges, we use four metrics to evaluate the LLM generated code, capturing code quality and physical accuracy of the solution, plus a holistic statistic, Success Rate. The details are as follows. **1) Executability** ($M_{\text{exec}}$): Similar to *CFDCodeBench*, we assign a value of 1 for successful execution of OpenFOAM using LLM generated case files and 0 otherwise. **2) Folder and File Structure** ($M_{\text{struct}}$): Generating the correct files and placing them in their respective folders is critical to the successful and accurate execution of the simulation workflow. The absence or misplacement of files can lead to failed execution of the case and/or inaccuracy of the generated output. Here, we use the ROUGE similarity metric (Lin, 2004) to compare the reference folder structure of the OpenFOAM cases with the LLM generated folder structure and provide a score between 0 and 1. **3) File Similarity** ($M_{\text{file}}$): This metric compares the content of the generated files with the reference OpenFOAM files using the ROUGE metric. **4) Relative Error** ($M_{\text{NMSE}}$): We use the same approach as *CFDCodeBench* Equation (1), comparing the LLM generated solution to a reference solution at the final time of the prescribed simulation window. **5) Success Rate**: We define Success Rate as the fraction of cases where just $M_{\text{exec}}$ and $M_{\text{NMSE}}$ achieves a score of 1.

## 4 EXPERIMENTS

In this section, we present results across a wide range of LLMs and agent frameworks that demonstrate the difficulty and realism of our benchmark.

### 4.1 EXPERIMENTAL SETUP

For benchmark tasks, we compare the performance of five closed-weight models including Claude Sonnet 3.5 (Anthropic, 2024), o3-mini (OpenAI, 2024b), Gemini 2.5 Flash (DeepMind, 2025), Claude Haiku 3.5 (Anthropic, 2024), and GPT-4o (OpenAI, 2024a), and one open-source model Gemma-2-9B-IT (Team, 2024). The temperature parameter is set to 0.0 for the models in evaluation in all experiments, except for o3-mini, which does not allow us to change the default temperature parameters and the value of this parameter is undisclosed. On *CFDQuery* and *CFDCodeBench*, LLMs use a standard zero-shot prompt template that describes the task and the output format. For *FoamBench*, we evaluate LLMs zero-shot, as well as with agentic frameworks (described next). We use OpenFOAM v10 for all experiments.

**Agentic frameworks for FoamBench**  Automating OpenFOAM using LLM is a complicated task, which we find benefits from agentic frameworks. Hence, for *FoamBench*, we not only compare various LLMs, but we also compare two agentic frameworks: *MetaOpenFoam* (Chen et al., 2024a) and *Foam-Agent* (Yue et al., 2025). Both of them assign agent roles for Retrieval-Augmented Generation (RAG) (Lewis et al., 2020) , file generation, running, and reviewing (Reviewer). These components enable the system to retrieve files from similar simulations to use as exemplars and to get intermediate feedback for re-attempting file generation if necessary. To assess the individual contributions of these components, we benchmark three configurations: (1) with RAG, with Reviewer; (2) with RAG, without Reviewer (3) without RAG, with Reviewer. The absence of RAG and Reviewer indicates zero-shot LLM prompting-based generation, which is used as a baseline to compare the improvements due to these agent roles.

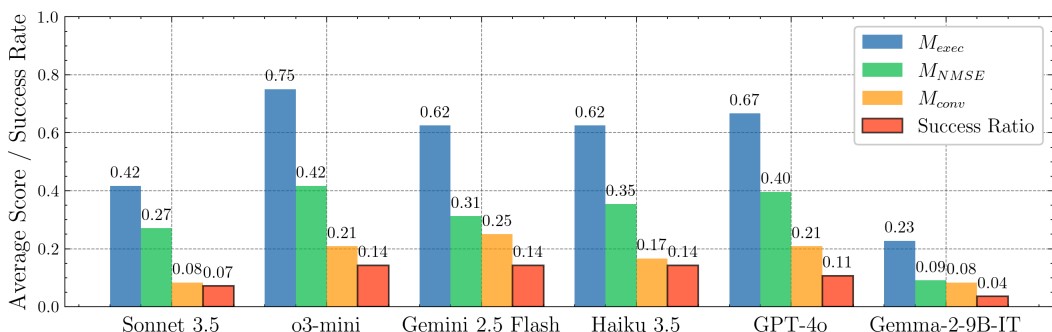

**Figure 4:** Average metric score and Success Rate for *CFDCodeBench*. The Success Rate for even the best performing models are around 14%, suggesting the challenging nature of the problems in this benchmark.

**Table 1:** Zero-shot prompt LLM performance with Sonnet 3.5 (best performing model) on *FoamBench Basic* and *Advanced*.

| Dataset | $M_{\text{exec}}$ | $M_{\text{struct}}$ | $M_{\text{file}}$ | $M_{\text{NMSE}}$ | *Success Rate* |
|---|---|---|---|---|---|
| FoamBench Basic | 0.064 | 0.670 | 0.506 | 0.050 | *0.045* |
| FoamBench Advanced | 0.017 | 0.773 | 0.573 | 0.009 | *0.007* |

## 4.2 RESULTS

The Success Rate of different models for the three benchmark tasks is shown in Figure 2. The *FoamBench* results are from the Foam-Agent framework, consisting of RAG and Reviewer, and using Sonnet 3.5, as this configuration yielded the strongest performance in our evaluations. Detailed *FoamBench* results are shown in Table 7. All closed-weight models perform well on *CFDQuery*, while the open sourced model could only answer 60% of the questions correctly. O3-mini performs the best in this task, which is not unexpected as it excels at logical reasoning and structured responses, producing 92% correct answers. On *CFDCodebench* and *FoamBench*, we see a drastic fall in Success Rate dropping to 14% in *CFDCodeBench* and 34% in *FoamBench Basic* and 25% in *FoamBench Advanced* for the best performing models. It is interesting to note that Sonnet 3.5 performs the best among other models by some margin in *FoamBench*, which is not seen in the other tasks. However, it costs higher per run on average ($6.56) than, e.g., GPT-4o ($0.42)-see Table 8.

**CFDCodeBench** Figure 4 illustrates the breakdown of metric scores and Success Rate as defined in Section 3.3 for different models. The accuracy and convergence metrics highlight the importance of holistic evaluation beyond syntactic correctness, which is often lacking in studies.

**FoamBench** Average metric scores and Success Rate of different models using the *Foam-Agent* framework with RAG and Reviewer is shown in Figure 5. Sonnet 3.5 was found to the best performing model for *FoamBench* tasks. The results of non-agentic zero-shot prompting with Sonnet 3.5 is provided in Table 1 to serve as a baseline for improvements due to the RAG and Reviewer roles (Table 2). This table also shows a comprehensive comparison between the two frameworks, MetaOpenFOAM and Foam-Agent, on *FoamBench* Basic and Advanced datasets. Detailed results on the impact of different models, framework and variations are provided in Appendix A.4.1.

## 5 DISCUSSION

**Importance of physical and numerical accuracy metrics** While all models demonstrate strong performance on *CFDQuery*—with Success Rate ranging from 60% (Gemma-2-9B-IT) to 92% (o3-mini), *performance significantly declines on tasks requiring physical and numerical accuracy*. To provide a holistic evaluation of model performance in *CFDCodeBench* and *FoamBench*, we reported multiple metrics and the stricter Success Rate. The latter aggregates success across code executability $M_{\text{exec}}$, numerical convergence $M_{\text{conv}}$, and physical accuracy $M_{\text{NMSE}}$, offering a practical view of model capabilities. From Figure 4, it is evident that most closed-weights models produce executable

**Table 2:** Component-wise mean scores and Success Rate for Claude Sonnet 3.5 on *FoamBench* Basic and Advanced, comparing MetaOpenFOAM vs. Foam-Agent.

| Dataset | Variation | MetaOpenFOAM | | | | | Foam-Agent | | | | |
|---|---|---|---|---|---|---|---|---|---|---|---|
| | | $M_{\text{exec}}$ | $M_{\text{struct}}$ | $M_{\text{file}}$ | $M_{\text{NMSE}}$ | *Success Rate* | $M_{\text{exec}}$ | $M_{\text{struct}}$ | $M_{\text{file}}$ | $M_{\text{NMSE}}$ | *Success Rate* |
| FoamBench Basic | RAG + Reviewer | 0.555 | 0.883 | 0.763 | 0.173 | *0.136* | 0.836 | 0.879 | 0.778 | 0.427 | *0.336* |
| | RAG + No Reviewer | 0.064 | 0.810 | 0.728 | 0.023 | *0.009* | 0.373 | 0.668 | 0.599 | 0.232 | *0.200* |
| | No RAG + Reviewer | 0.400 | 0.747 | 0.522 | 0.195 | *0.145* | 0.473 | 0.862 | 0.647 | 0.291 | *0.245* |
| FoamBench Advanced | RAG + Reviewer | 0.125 | 0.775 | 0.599 | 0.125 | *0.125* | 0.625 | 0.792 | 0.621 | 0.406 | *0.250* |
| | RAG + No Reviewer | 0.000 | 0.743 | 0.594 | 0.000 | *0.000* | 0.188 | 0.771 | 0.609 | 0.156 | *0.125* |
| | No RAG + Reviewer | 0.375 | 0.655 | 0.451 | 0.344 | *0.187* | 0.250 | 0.806 | 0.592 | 0.188 | *0.125* |

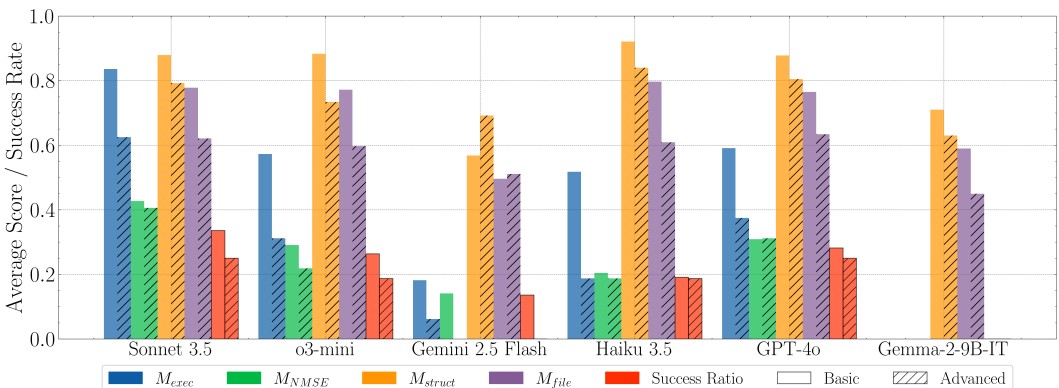

**Figure 5:** Average metric score and Success Rate for different models on *FoamBench* using *Foam-Agent* framework with RAG and reviewer. The Success Rate for even the best performing model (Sonnet 3.5) is 34% in basic dataset and 25% in the advanced dataset.

Python code in over 60% of cases, but these numbers are significantly worse for physical and numerical accuracy. For instance, in *FoamBench Basic*, the best Foam-Agent (Table 2) achieves good coding metrics $M_{\text{exec}} = 0.836$, $M_{\text{struct}} = 0.879$, $M_{\text{file}} = 0.778$, but the Success Rate is only 34% because of low physical accuracy. We see that the LLMs often fail to fully understand the prompts and lack domain-specific reasoning required to correctly apply fundamental CFD concepts—such as flux discretization schemes, appropriate time integration strategies, and consistent boundary treatments. This highlights a critical gap in current models' capabilities when it comes to generating reliable and physically consistent CFD code.

**Zero-shot prompting for OpenFOAM** Zero-shot prompting produces close to 0% Success Rate even for the best performing model (Sonnet 3.5) as shown in Table 1, highlighting the need for agentic frameworks when it comes to running OpenFOAM. For example, it is difficult for current LLMs to produce all of the required input files in a zero-shot manner. Even with prompt engineering the increment in success rate under zero-shot setting is only marginal (0.007 to 0.012), for Claude Sonnet 3.5 with further details in Appendix A.4.4. We observe that Sonnet 3.5 and o3-mini (Appendix A.4.1) have the most successful zero-shot runs.

**Role of RAG and Reviewer** RAG provides the framework with similar simulation files and the Reviewer allows for a trial and error approach to running OpenFOAM cases, mimicking human troubleshooting. The absence of either decreases the Success Rate by approximately 10% (Table 2), underscoring their critical roles in achieving optimal performance within the proposed framework.

**Spatial reasoning** The CFD simulation workflows in *FoamBench* have preprocessing steps where a correct geometry and mesh file must be generated by the LLM. To handle real-world workflows, LLMs should be able to extrapolate to novel geometries. We highlight a particular case from *FoamBench Advanced*, doubleSquare, which is an incompressible flow over two square obstacles. The geometry produced by the *Foam-Agent*, in comparison to the reference geometry, is visualized in

Figure 6. The prompt clearly defines the location of the obstacles, but the lack of spatial reasoning capabilities in LLMs appears to produce an incorrect geometry and mesh. We highlight that the ability of LLMs to understand geometry is a major area in need of improvement.

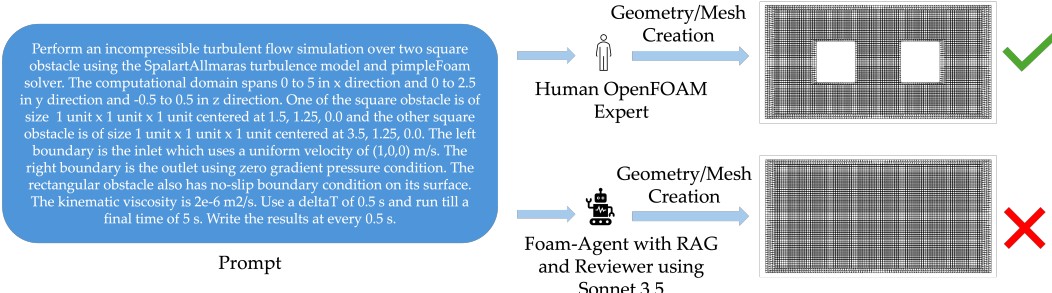

**Figure 6:** Comparison of the geometry and mesh generated by the *Foam-Agent* (Yue et al., 2025) (RAG and Reviewer) with Sonnet 3.5 for the doubleSquare case against human expert.

## 6    CONCLUSION

In this work, we introduced *CFDLLMBench*, the first benchmark to holistically evaluate graduate-level knowledge, numerical and physical reasoning, and practical simulation capabilities of LLMs for CFD. We accomplish this by structuring the benchmark into three progressively challenging tiers, namely, *CFDQuery*, *CFDCodeBench*, and *FoamBench*. Our results highlight both the promise and the current limitations of LLMs in solving advanced scientific workflow automation problems, which require software expertise such as tool-calling and long-context understanding, as well as accurate physical modeling. We expect that *CFDLLMBench* will serve as a valuable testbed for advancing LLM capabilities in scientific computing, and encourage future work on domain-grounded, execution-based benchmarks across other areas of science and engineering.

## 7    REPRODUCIBILITY STATEMENT

All problems in our benchmark were collected from open, publicly available sources or were authored specifically for this benchmark. Accordingly, *CFDLLMBench* is released under the terms of the BSD 3-Clause License, making it free to use, modify, and redistribute, including for commercial purposes, provided that the license conditions are met. Our benchmark pipeline relies exclusively on free and open-source software, ensuring that it is accessible to all users without the need for paid subscriptions. Furthermore, we release not only the dataset (prompts), but also the complete codebase, fully containerized with Docker, to enable reproducibility. The code to run the benchmark is attached as supplementary material and can also be found at https://anonymous.4open.science/r/cfdllmbench-5654. The code repository also provides instruction on how to run the benchmarks and the anonymous private links to the dataset being used. This comprehensive release allows future researchers to easily utilize, reproduce, or extend our benchmark with minimal overhead.

## 8    ETHICS STATEMENT

While the nature of the human work in this study did not warrant formal Institutional Review Board (IRB) review, we nevertheless followed all ethical norms and standards of the host academic institution when performing the human tasks associated with dataset creation. All human experts involved were members of the research project and were fairly compensated for their time and expertise. No personally identifiable or sensitive data are included in the released dataset, and all data sources are either public or used with appropriate permissions. Large language models were used solely for minor English editing and grammar polishing. These tools were not involved in the conception, design, execution, analysis, or interpretation of the research.

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

## A  APPENDIX

### A.1  HUMAN BASELINE INFERENCE

Establishing appropriate human references is non-trivial because performance depends strongly on the evaluator's domain knowledge and experience, which are difficult to standardize and quantify. A comprehensive human study would require recruiting domain experts, designing controlled protocols, and measuring expert effort and accuracy, an undertaking that is substantial in both organization and time. Nevertheless, approximate human performance can be reasonably inferred from established expectations of CFD practitioners, and these comparisons help to contextualize our results:

- **CFDQuery (Conceptual Knowledge).** Leading closed-source LLMs achieve 85–90% accuracy on graduate-level CFD questions, a level comparable to or exceeding what a typical CFD engineer would attain in a closed-book setting, where humans usually rely on references for such broad coverage.

- **CFDCodeBench (Numerical Reasoning and Code Generation).** The top LLM scores only 14% on simple PDE solver tasks (e.g., diffusion, Burgers). A CFD trained graduate student can reliably solve these with high accuracy by writing a small script or reusing existing templates, highlighting the gap between LLM memorization and genuine reasoning/coding.

- **FoamBench (Workflow Automation).** Even with an agentic setup, the best model achieves only 34% success on standard OpenFOAM tutorial cases. A CFD engineer familiar with OpenFOAM would easily solve most of these tasks, showing that current LLMs struggle with decomposition and physics-driven workflow generation.

Future iterations of the benchmark may incorporate formal human baselines to enable a more direct quantitative comparison between human and model performance.

## A.2   NMSE THRESHOLDS

The NMSE threshold used in this (lower bound of 10% and upper bound of 30%) were not chosen arbitrarily but are grounded in engineering practice and further supported by an empirical sensitivity analysis.

**A.2.0.1   Engineering Practice.**   CFD engineering practice commonly follows the thumb rule that an NMSE below 10% indicates an accurate simulation, while errors above 30% mark the upper limit for accuracy. In CFD and related engineering fields, an NMSE (or relative error) below approximately 10%, typically resulting from well-configured numerical setups, is widely regarded as indicative of an accurate and reliable simulation. Conversely, errors exceeding 30% are generally considered practically unacceptable when validating simulations against numerical ground truth. These brackets are routinely used in both academic validation studies and industrial verification.

**A.2.0.2   Empirical Sensitivity Analysis.**   To further justify our choice, we conducted a sensitivity study by varying the thresholds and observing their effect on both mean NMSE score and the true success rate.

**Table 3:** Mean NMSE scores with varying lower bounds (upper bound fixed at 30%).

| Lower Bound | Mean NMSE Score |
|---|---|
| 1% | 0.3909 |
| 5% | 0.4000 |
| 10% | 0.4273 |
| 15% | 0.4318 |

**Table 4:** Sensitivity of true success rate to different lower NMSE cutoffs (upper bound fixed at 30%).

| Lower NMSE Bound | True Success Rate |
|---|---|
| 1% | 26.4% |
| 5% | 28.2% |
| 10% | 33.6% |
| 15% | 34.5% |

The tables show a clear progression, with the strongest gain observed at 10%, beyond which the increase is marginal.

**Table 5:** Mean NMSE scores with varying upper bounds (lower bound fixed at 10%).

| Upper Bound | Mean NMSE Score |
|---|---|
| 0.25 | 0.4045 |
| 0.30 | 0.4273 |
| 0.40 | 0.4955 |
| 0.45 | 0.5045 |

It can be seen that beyond 30%, the metric becomes overly accommodative and can include edge cases. The combination of domain-standard brackets (10% and 30%) and our sensitivity analysis

demonstrates that 10% is the optimal cutoff for accurately identifying correct simulations, while 30% serves as a natural upper limit for defining unacceptable solutions. These thresholds align with established CFD practices and ensure that the metric remains interpretable and meaningful.

## A.3 DATASET CURATION

### A.3.1 CFDQUERY

This Question and Answer dataset spans a broad spectrum of PDEs, numerical methods and error-analysis topics. It delves into classical finite difference and finite volume schemes applied to 1D advection, 1D diffusion, 1D Burgers equation, etc (e.g. modified-equation analyses of central-difference+RK3, Lax–Friedrichs dissipation, upwind bias) and proceeding through high-order stencils (fourth-order central, compact schemes, WENO) and their dispersion/dissipation properties. It also involves questions based on non-uniform and curvilinear grids—deriving coefficient formulas for second derivatives on unequal spacings, analyzing truncation errors on non-orthogonal meshes, and enforcing the geometric-conservation law. Further the questions probe multi-dimensional flows (Poisson, Navier–Stokes channel and cavity, Rayleigh–Bénard, KdV–Burgers) with questions on stability criteria, and leading-order error terms. Finally, the set includes advanced topics in high-order discontinuous-Galerkin.

These questions are curated, reviewed and solved by human CFD experts before adding to the dataset. The sources of the dataset include textbooks and online sources. Each question is self-sufficient and provides four options to the LLM to select the right answer from. In addition we also provide a system prompt to the LLM to assign the role it will be playing when answering the questions. The system prompt is given below.

> **Prompt**
>
> ```
> You are an expert computational fluid dynamics researcher. For each multiple-choice
> question, read the question and its four options, then respond with only the number
> (1, 2, 3, or 4) corresponding to the correct answer.
> ```

### A.3.2 CFDCODEBENCH

To construct *CFDCodeBench*, we curated a dataset of 24 computational fluid dynamics (CFD) problems from publicly available GitHub repositories and established numerical solver packages. Foundational problems were selected from the widely used *CFD Python: the 12 Steps to Navier-Stokes Equations* repository (Barba & Forsyth, 2018) and other educational sources such as *ENGR 491 - Computational Fluid Dynamics* (Lab, 2024). These 17 problems, which include well-documented tutorials and reference code. To introduce more challenging scenarios, we incorporated 7 advanced problems from the Dedalus Project (Burns et al., 2020), which offers flexible PDE solvers based on spectral methods. These problems lacked detailed tutorials, so CFD experts reviewed the source code and authored accompanying descriptions. All problem descriptions and corresponding solutions were manually validated to ensure correctness and consistency.

- **1D Burgers Equation:** Simulates 1D viscous Burgers equation with Dirichlet boundaries.

- **1D Diffusion Equation:** Models scalar diffusion over time with piecewise constant initial conditions in a 1D domain.

- **Euler's equation for compressible flow in a shock tube:** Simulates shock propagation in a 1D shock tube using the Euler equations with reflective boundaries. The solution to this equation is highly susceptible to numerical instabilities.

- **1D linear convection Equation:** Solves undamped linear convection of a Gaussian wave with periodic boundaries.

- **1D non-linear convection Equation:** Captures nonlinear wave propagation with sinusoidal initial conditions and periodic boundaries.

- **2D Burgers Equation:** Simulates viscous flow in both x and y directions using the 2D Burgers' equation with Dirichlet boundaries.

- **2D Convection Equation:** Models 2D inviscid convection of a velocity disturbance with constant boundary conditions.

- **2D Diffusion Equation:** Solves a 2D scalar diffusion problem with fixed values on all boundaries and an initial high-temperature patch.

- **2D inviscid Burgers Equation:** Captures shock formation in a 2D inviscid Burgers' flow using a square domain and periodic boundaries.

- **2D Laplace Equation:** Solves a steady-state potential problem with mixed Dirichlet and Neumann conditions.

- **2D Linear Convection Equation:** Simulates scalar convection in two directions from a localized initial disturbance.

- **2D Navier-Stokes equation in a cavity:** Computes incompressible viscous flow in a lid-driven cavity setup using the Navier-Stokes equations.

- **Channel Flow with Navier–Stokes:** Solves channel flow with periodic inlet/outlet, no-slip top/bottom, and constant body force.

- **2D Poisson Equation:** Solves the 2D Poisson equation with localized sources and Dirichlet boundaries.

- **2D Steady Heat Equation:** Models steady-state heat conduction on a rectangular plate with fixed temperatures on all edges.

- **2D Unsteady Heat Equation:** Simulates time-dependent heating with a Gaussian source term and fixed boundaries.

- **Fully-developed turbulent flow in a channel:** Uses a Cess turbulence model to compute velocity profiles in a turbulent channel with effective viscosity.

- **1D Korteweg-de Vries / Burgers Equation:** Models the combined effects of diffusion and dispersion in wave dynamics using the KdV-Burgers equation.

- **2D horizontally-periodic Rayleigh-Benard convection Equation:** Simulates buoyancy-driven convection with temperature gradients and periodic lateral boundaries.

- **2D periodic incompressible shear flow with a passive tracer field:** Models shear flow evolution and passive tracer transport with periodic boundaries.

- **Flow past circular cylinder:** Simulates vortex shedding behind a cylinder using streamfunction-vorticity formulation in polar coordinates.

- **Lane-Emden Equation:** Solves a spherically symmetric nonlinear Poisson equation used in astrophysics.

- **Incompressible Navier Stokes equations in a lid-driven cavity:** Captures recirculating flow in a square cavity driven by a moving top wall.

- **Linear stability eigenvalue problem for pipe flow:** Solves the linear stability eigenvalue problem of pipe flow using the linearized Navier–Stokes equations.

### A.3.2.1 Prompt Design and Methodology for CFDCodeBench

**Structured Prompt Generation**   We adopt a structured, JSON-to-natural language pipeline for prompt generation. Each PDE problem is described in a JSON object with fields such as:

- `equation`: The governing PDE, formatted using LaTeX, e.g., $\frac{\partial u}{\partial t} + u\frac{\partial u}{\partial x} = \nu\frac{\partial^2 u}{\partial x^2}$.

- `boundary conditions`: A description of boundary behavior, written in either LaTeX or plain text, e.g., periodic boundary conditions such as $u(0) = u(2\pi)$.

- `initial conditions`: The initial state of the solution field, typically in compact LaTeX format, e.g., $u(x,0) = \begin{cases} 2, & \text{if } 0.5 \le x \le 1 \\ 1, & \text{otherwise} \end{cases}$.

- `domain`: The spatial and temporal domain of the problem, for instance, $x \in [0, 2\pi]$, $t \in [0, 0.14\pi]$.

- **save values**: A list of solution variables (e.g., $u$, $v$, $p$) that should be saved at the final time step.

- **numerical method** (optional): Specifies the numerical scheme to be used, e.g., finite difference method (FDM), finite volume method (FVM), or finite element method (FEM). This field may be omitted when using FDM as the default method.

**Example Problem Description in JSON Format**

---

**Problem Description (JSON Format)**

```
{
    "1D_Burgers_Equation": {
        "equation": "\\[\n  \\frac{\\partial u}{\\partial t} + u \\
            frac{\\partial u}{\\partial x} = \\nu \\frac{\\partial^2 u
            }{\\partial x^2}\n\\]\n\nwhere:\n- \\( u(x,t) \\) is the
            velocity field\n- \\( \\nu = 0.07 \\) is the viscosity
            coefficient\n- \\( x \\) is the spatial coordinate\n- \\(
            t \\) is time",
        "boundary conditions": "Periodic boundary conditions:\n\\[\n
            u(0) = u(2\\pi)\n\\]",
        "initial conditions": "\\[\n  u = -\\frac{2\\nu}{\\phi} \\frac
            {\\partial \\phi}{\\partial x} + 4\n\\]\nwhere:\n\\[\n  \\
            phi = \\exp\\left(\\frac{-x^2}{4\\nu}\\right) + \\exp\\
            left(\\frac{-(x - 2\\pi)^2}{4\\nu}\\right)\n\\]",
        "domain": "- Spatial domain: \\( x \\in [0, 2\\pi] \\), -
            Temporal domain: (t \\in [0, 0.14\\pi])",
        "save values": "u",
        "numerical method": "finite difference method"
    },
}
```

---

**Prompt Generation Function**

Following the construction of the problem description in JSON format, we systematically generate structured user prompts for the LLM based on the provided information. The conversion from JSON to natural language is automated through a prompt generation function, which formats the fields (e.g., equation, boundary conditions, initial conditions, domain, save values, and numerical method) into a coherent and standardized instruction. This structured prompt explicitly communicates the problem setup and expected outputs, ensuring consistency across different tasks and minimizing ambiguity during code generation. The generated prompts follow a fixed template to guarantee reproducibility and comparability throughout the benchmark.

The prompt is designed to achieve the following goals:

**Deterministic and parseable:** The generated prompts follow a consistent structure, enabling easy parsing and reproducibility.

**Clear separation of problem components:** Each prompt explicitly isolates the problem definition, including the partial differential equation (PDE), boundary conditions (BC), initial conditions (IC), and domain specifications.

**Specification of code generation requirements:** Prompts clearly define additional requirements such as the numerical method, output format, and variables to be saved.

**Solver-agnostic design:** While prompts recommend a numerical method (e.g., finite difference method (FDM)), they remain flexible and do not enforce dependence on a particular solver framework.

**Prompt Generation Function**

```python
def generate_prompt(data):
    parts = [
        "You are given the following partial differential equation (PDE) problem:\n",

        "**Equation:**\n" + data.get("equation", "") + "\n",
        "**Boundary Conditions:**\n" + data.get("boundary conditions", "") + "\n",
        "**Initial Conditions:**\n" + data.get("initial conditions", "") + "\n",
        "**Domain:**\n" + data.get("domain", "") + "\n",
        "**Numerical Method:**\n" + data.get("numerical method", "") + "\n"
    ]

    # Check for 'save_values' and add to task description
    save_values = data.get("save_values", [])
    save_values_str = ", ".join(save_values) if save_values else "the relevant
        variables specified for the problem"
    # Always end with task specification for the code
    parts.append(
        "### Task:\n"
        "- Write Python code to numerically solve the given CFD problem. Choose an
            appropriate numerical method based "
        "on the problem characteristics.\n"
        "- If the problem is **unsteady**, only compute and save the **solution at
            the final time step**.\n"
        "- For each specified variable, save the final solution as a separate '.npy'
            file using NumPy:\n"
        " - For **1D problems**, save each variable as a 1D NumPy array.\n"
        " - For **2D problems**, save each variable as a 2D NumPy array.\n"
        "- The '.npy' files should contain only the final solution field (not
            intermediate steps) for each of the "
        "specified variables.\n"
        "- **Save the following variables** at the final time step:\n"
        + save_values_str + "\n"
                        "(Each variable should be saved separately in its own '.npy
                            ' file, using the same name as "
                        "provided in 'save_values').\n"
                        "- Ensure the generated code properly handles the solution
                            for each specified variable "
                        "and saves it correctly in '.npy' format.\n"
                        "- **Return only the complete, runnable Python code** that
                            implements the above tasks, "
                        "ensuring no extra explanations or information is included.
                            "
    )

    return "\n".join(parts)
```

**Example Generated User Prompt**

**Generated User Prompt**

```
{
    "1D_Burgers_Equation": "You are given the following partial
        differential equation (PDE) problem:\n\n**Equation:**\n\\[\n
        \\frac{\\partial u}{\\partial t} + u \\frac{\\partial u}{\\
        partial x} = \\nu \\frac{\\partial^2 u}{\\partial x^2}\n\\]\n\
        nwhere:\n- \\( u(x,t) \\) is the velocity field\n- \\( \\nu =
        0.07 \\) is the viscosity coefficient\n- \\( x \\) is the
        spatial coordinate\n- \\( t \\) is time\n\n**Boundary
        Conditions:**\nPeriodic boundary conditions:\n\\[\n  u(0) = u(
        2\\pi)\n\\]\n\n**Initial Conditions:**\n\\[\n  u = -\\frac{2\\
        nu}{\\phi} \\frac{\\partial \\phi}{\\partial x} + 4\n\\]\
        nwhere:\n\\[\n  \\phi = \\exp\\left(\\frac{-x^2}{4\\nu}\\right
        ) + \\exp\\left(\\frac{-(x - 2\\pi)^2}{4\\nu}\\right)\n\\]\n\n
        **Domain:**\n- Spatial domain: \\( x \\in [0, 2\\pi] \\), -
        Temporal domain: (t \\in [0, 0.14\\pi])\n\n**Numerical Method:
        **\nfinite difference method\n\n### Task:\n- Write Python code
         to numerically solve the given CFD problem. Choose an
        appropriate numerical method based on the problem
        characteristics.\n- If the problem is **unsteady**, only
        compute and save the **solution at the final time step**.\n-
        For each specified variable, save the final solution as a
        separate '.npy' file using NumPy:\n  - For **1D problems**,
        save each variable as a 1D NumPy array.\n  - For **2D problems
        **, save each variable as a 2D NumPy array.\n- The '.npy'
        files should contain only the final solution field (not
        intermediate steps) for each of the specified variables.\n- **
        Save the following variables** at the final time step:\nthe
        relevant variables specified for the problem\n(Each variable
        should be saved separately in its own '.npy' file, using the
        same name as provided in 'save_values').\n- Ensure the
        generated code properly handles the solution for each
        specified variable and saves it correctly in '.npy' format.\n-
         **Return only the complete, runnable Python code** that
        implements the above tasks, ensuring no extra explanations or
        information is included.",
}
```

**System Prompt**  In addition to the user prompt, we employ a fixed system prompt to explicitly define the role of the LLM. For models that do not support system prompts natively, the system prompt is appended to the beginning of the user prompt to ensure consistent behavior across different models.

**System Prompt**

```
"You are a highly skilled assistant capable of generating Python code
    to solve CFD problems "
                    "using appropriate numerical methods."
                    "Given the problem description, you should reason
                        through the problem and determine the best "
                    "approach for discretizing and solving it,"
                    "while respecting the specified boundary conditions
                        , initial conditions, and domain.\n"
                    "For unsteady problems, save only the solution at
                        the final time step. For 1D problems, "
                    "save a 1D array; for 2D problems, save a 2D array
                        .\n"
                    "Ensure the code follows the user's specifications
                        and saves the requested variables exactly "
                    "as named in 'save_values'.\n"
                    "Your task is to generate the correct, fully
                        runnable Python code for solving the problem "
                    "without additional explanations."
```

**Task Execution Protocol**   We evaluate code generation performance across a range of large language models (LLMs). When configurable, we set the decoding temperature to 0 to minimize randomness and encourage deterministic outputs. For models where temperature is fixed by the provider, we use the default setting. We do not explicitly constrain the maximum number of generated tokens; however, we note that some models impose an internal context window limit of approximately 8000 tokens, which encompasses both prompt and output. No additional stop sequences or length truncation strategies are applied.

**Execution and Output Validation**   For each generated response, we extract the Python code and execute it in a controlled environment. The resulting numerical solution is saved as a NumPy array and compared against the expert-provided reference solution. To quantify the accuracy of the generated results, we compute the *Normalized Mean Squared Error* (NMSE) between the predicted and true solution arrays. In cases where the shapes of the predicted and reference arrays do not match, we apply interpolation to align the dimensions before comparison. Additionally, we visualize both the predicted and reference solutions as images to qualitatively assess agreement and identify structural discrepancies. All code execution is sandboxed with timeouts (default to be 60 seconds) to prevent infinite loops or excessive resource usage.

### A.3.3   FOAMBENCH

This class of benchmark study focuses on OpenFOAM dataset. Being an open source framework, there are multitudes of cases available. However, scraping through all such available cases to generate the dataset can create additional challenges in evaluating the effectiveness of such frameworks. Also, OpenFOAM is capable of simulating complex geometries that we see in real life scenarios (e.g. flow over an airplane or automobile), which requires creation of a CAD geometry outside of OpenFOAM using specialized tools and further creating a computational mesh, which is then imported into OpenFOAM for further CFD analysis. In the current benchmark, we want to evaluate an end to end usage of OpenFOAM, where it can generate its own geometry and mesh and further do the required numerical analysis. Hence we stick with geometries that can be easily described using natural language and/or part of the tutorial. With these guidelines in mind we curate *FoamBench Basic* and *FoamBench Advanced* dataset.

**A.3.3.1   FoamBench Basic**   The basic dataset consists of tutorial problems with regards to the physics, geometry and models. We picked out 11 different tutorial problems namely:

**BernardCells**   Simulates Rayleigh-Bénard convection within a rectangular cavity, driven by a temperature gradient between the hot bottom wall and the cold top wall. It models buoyancy-driven flow and thermal instabilities using Boussinesq approximation.

**Cavity**   Classic lid-driven cavity problem with a square geometry and a moving top wall. It is used to validate laminar incompressible solvers and study vortex formation.

**counterFlowFlame2D**   Models a 2D counter-flow diffusion flame with detailed combustion and chemistry. The domain consists of opposing inlets where fuel and oxidizer meet, ideal for flame structure studies.

**Cylinder**   Simulates flow past a stationary cylinder in a 2D channel. Demonstrates vortex shedding and drag, and is widely used for benchmarking turbulence models.

**damBreakWithObstacle**   A multiphase VOF (Volume of Fluid) simulation of a dam break in the presence of a central obstacle. Tests free-surface dynamics and wave-obstacle interactions.

**forwardStep**   Compressible flow over a sudden forward-facing step in a duct. Used to observe shock reflections, expansion fans, and flow separation at high Mach numbers.

**obliqueShock**   This case simulates compressible, inviscid supersonic flow over a flat domain, leading to the formation of an oblique shock wave. Unlike classic textbook setups that use a physical wedge to induce the shock, this case creates an oblique shock by imposing different velocity and temperature conditions at the inlet boundary of a flat channel.

**pitzDaily**   Simulates turbulent flow in a channel with a backward-facing step. It is a standard test case for turbulence model validation due to its separation and reattachment zones.

**shallowWaterWithSquareBump**   Uses the shallow water equations to model surface flow over a square bump. Tests numerical schemes for water surface deformation and hydraulic jumps.

**squareBend**   Involves internal flow through a 90-degree square bend. Demonstrates secondary flow effects caused by pressure gradients in curved channels.

**wedge**   An axisymmetric setup often used for supersonic flow around wedges. Useful for studying inviscid compressible flows with symmetry assumptions.

We create 10 variations for each of these 11 datasets by varying parameters such as inlet velocity, viscosity, boundary temperature values etc. This gives us 110 distinct OpenFOAM case files which can be used as reference. The benchmark dataset consists of reference human made OpenFOAM case files and a prompt for the agentic framework describing the problem to be solved. We provide an example from our basic benchmark dataset for the forwardStep case. The reference folder structure in which the files are to be organized is shown below.

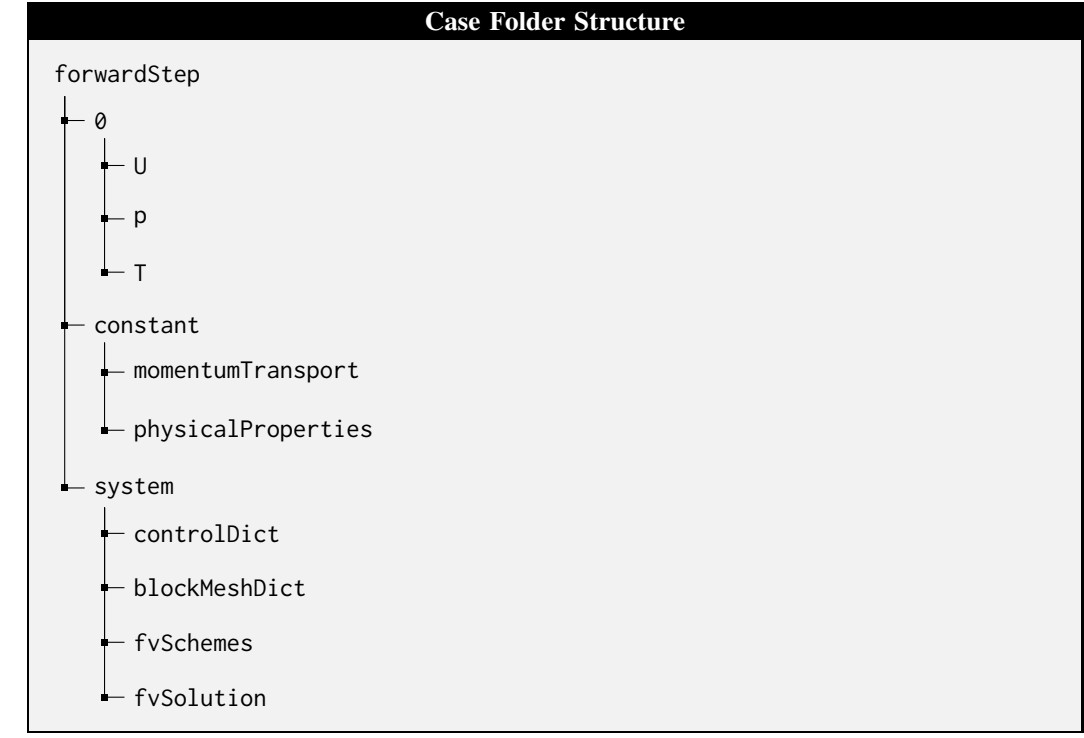

We show the details of these files which is used as reference in Figure 7, Figure 8 and Figure 9

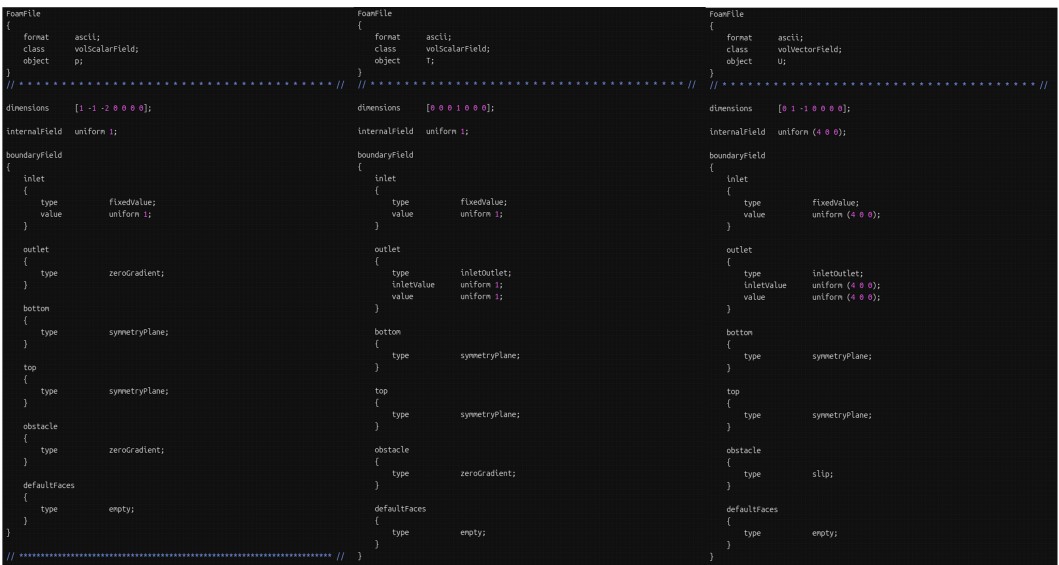

**Figure 7:** OpenFoam reference case files defining the initial and boundary conditions.

Finally the prompt that is input to the frameworks is shown below. Since they are tutorial problems, we do not describe the geometry in great lengths and assume the RAG should be able to pick it out based on the description.

**Figure 8:** OpenFoam reference case files defining the physical propereties and turbulence models.

**Figure 9:** OpenFoam reference case files defining the solver configurations, geometry and mesh.

> **Prompt**
>
> ```
> Do a laminar, compressible flow over a forward-facing step using the rhoCentralFoam
> solver. Boundary conditions include a fixed velocity of 3 m/s and temperature of 1K
> at the inlet and slip conditions on the obstacle. Use a timestep of 0.002 and output
> every 0.1. Final time is 4.
> ```

**A.3.3.2  FoamBench Advanced**   The advanced dataset consists of cases that are not part of the tutorial. These cases are used to evaluate the LLMs capabilities in piecing together the information from available tutorials and extrapolating to major changes that the user requests in the following categories:

- Turbulence Model Changes: Changes in turbulence models requires the LLM to understand the required file changes and parameter changes. Shifting from one turbulence model to another not only requires simple option changes in files but may also involve additional file to be created in the initial and boundary condition folders specifying the parameters for the given turbulence model.

- Geometric Modifications: We ask the LLM to make changes to the geometry in tutorial problems in changing the size of the domain. This requires the LLM to understand spatial configuration of a given problem and make the required changes to the domain configuration in the geometry definition.

- Unseen Geometry: In these tasks we ask LLM to create new obstacle shapes within the flow domain. Such tasks can be quite complex in nature, requiring the LLM to understand the new geometric requirements of the user and piece together information from its own knowledge and tutorial cases and perform sptaial reasoning to generate these geometries.

We have curated a total of 16 cases, covering the above mentioned aspects of extrapolation. A sample prompt given as the input to the framework is shown below

> **Prompt**
>
> ```
> Perform an incompressible turbulent flow simulation over a 2D diamond obstacle using
> the k-epsilon RANS turbulence model and pimpleFoam solver. The computational domain
> spans 0 to 15 in x direction and 0 to 5 in y direction and -0.5 to 0.5 in z direction.
> The diamond obstacle is a square rotated by 45 degrees with diagonal length of 1 unit
> centered at 2.5 x 2.5 x 0.0. Use one cell in z direction making the geometry effectively
> 2D. Refine the mesh near to diamond. Use sufficient grid points to discretize the
> domain and dont use more than 10000 cells in your mesh. The left boundary is the
> inlet which uses a uniform velocity of (1,0,0) m/s. The right boundary is the outlet
> using zero gradient pressure condition. Top and bottom boundaries are fixed walls with
> nop-slip condition. The front and back faces are empty. The diamond obstacle also has
> no-slip boundary condition on its surface. The kinematic viscosity is 2e-6 $m^2$/s. Use
> a deltaT of 0.5 s and run till a final time of 5 s. Write the results at every 0.5 s.
> Use a maximum Courant number of 1.0.
> ```

Unlike *FoamBench Basic* these are unseen geometries. Hence the prompt is made descriptive enough for the LLM to understand the user need and generate the relevant blockMeshDict file containing the geometric details and mesh information. Since it is difficult to perform mesh control over natural language we only specify the maximum number of cells to be used and required refinement near the obstacle.

## A.4   AGENTIC FRAMEWORK

### A.4.1   DETAILED PERFORMANCE COMPARISON

**Table 6:** Performance of **Zero Shot Pure LLM** on *FoamBench Basic* and *FoamBench Advanced*.

| Variation | Model | | | | | | | | | | |
|---|---|---|---|---|---|---|---|---|---|---|---|
| | | \multicolumn{5}{c}{*FoamBench Basic*} | | | \multicolumn{5}{c}{*FoamBench Advanced*} | | | |
| | | $M_{\text{exec}}$ | $M_{\text{struct}}$ | $M_{\text{file}}$ | $M_{\text{NMSE}}$ | Success Rate | $M_{\text{exec}}$ | $M_{\text{struct}}$ | $M_{\text{file}}$ | $M_{\text{NMSE}}$ | Success Rate |
| | Sonnet 3.5 | 0.064 | 0.670 | 0.506 | 0.050 | 0.045 | 0.017 | 0.773 | 0.573 | 0.009 | 0.007 |
| | o3-mini | 0.009 | 0.788 | 0.529 | 0.009 | 0.009 | 0.000 | 0.707 | 0.408 | 0.000 | 0.000 |
| Zero Shot | Gemini 2.5 Flash | 0.000 | 0.828 | 0.573 | 0.000 | 0.000 | 0.000 | 0.666 | 0.406 | 0.000 | 0.000 |
| Pure LLM | Haiku 3.5 | 0.000 | 0.905 | 0.629 | 0.000 | 0.000 | 0.000 | 0.801 | 0.492 | 0.000 | 0.000 |
| | GPT-4o | 0.000 | 0.819 | 0.589 | 0.000 | 0.000 | 0.000 | 0.735 | 0.466 | 0.000 | 0.000 |
| | Gemma-2-9B-IT | 0.000 | 0.735 | 0.460 | 0.000 | 0.000 | 0.000 | 0.670 | 0.390 | 0.000 | 0.000 |

**Table 7:** Component-wise mean scores and true-Success Rates for each model and framework on *FoamBench* Basic (top) and Advanced (bottom).

| Variation | Model | \multicolumn{5}{c}{**MetaOpenFOAM**} | | | | | \multicolumn{5}{c}{**Foam-Agent**} | | | | |
|---|---|---|---|---|---|---|---|---|---|---|---|
| | | $M_{\text{exec}}$ | $M_{\text{struct}}$ | $M_{\text{file}}$ | $M_{\text{NMSE}}$ | Success Ratio | $M_{\text{exec}}$ | $M_{\text{struct}}$ | $M_{\text{file}}$ | $M_{\text{NMSE}}$ | Success Ratio |
| | Sonnet 3.5 | 0.555 | 0.883 | 0.763 | 0.173 | 0.136 | 0.836 | 0.879 | 0.778 | 0.427 | 0.336 |
| RAG | o3-mini | 0.491 | 0.872 | 0.664 | 0.236 | 0.227 | 0.573 | 0.883 | 0.772 | 0.291 | 0.264 |
| + | Gemini 2.5 Flash | 0.245 | 0.841 | 0.695 | 0.091 | 0.082 | 0.182 | 0.568 | 0.496 | 0.141 | 0.136 |
| Reviewer | Haiku 3.5 | 0.218 | 0.845 | 0.701 | 0.095 | 0.091 | 0.518 | 0.921 | 0.797 | 0.205 | 0.191 |
| | GPT-4o | 0.173 | 0.801 | 0.715 | 0.105 | 0.091 | 0.591 | 0.878 | 0.765 | 0.309 | 0.282 |
| | Gemma-2-9B-IT | 0.000 | 0.690 | 0.540 | 0.000 | 0.000 | 0.000 | 0.710 | 0.590 | 0.000 | 0.000 |
| | Sonnet 3.5 | 0.064 | 0.810 | 0.728 | 0.023 | 0.009 | 0.373 | 0.668 | 0.599 | 0.232 | 0.200 |
| RAG | o3-mini | 0.055 | 0.823 | 0.651 | 0.027 | 0.027 | 0.436 | 0.837 | 0.744 | 0.273 | 0.255 |
| + | Gemini 2.5 Flash | 0.009 | 0.793 | 0.682 | 0.009 | 0.009 | 0.191 | 0.811 | 0.685 | 0.145 | 0.136 |
| No Reviewer | Haiku 3.5 | 0.055 | 0.806 | 0.708 | 0.027 | 0.027 | 0.182 | 0.915 | 0.799 | 0.100 | 0.091 |
| | GPT-4o | 0.045 | 0.796 | 0.710 | 0.018 | 0.018 | 0.455 | 0.843 | 0.738 | 0.286 | 0.255 |
| | Gemma-2-9B-IT | 0.000 | 0.680 | 0.540 | 0.000 | 0.000 | 0.000 | 0.720 | 0.590 | 0.000 | 0.000 |
| | Sonnet 3.5 | 0.400 | 0.747 | 0.522 | 0.195 | 0.145 | 0.473 | 0.862 | 0.647 | 0.291 | 0.245 |
| No RAG | o3-mini | 0.045 | 0.623 | 0.347 | 0.000 | 0.000 | 0.009 | 0.811 | 0.549 | 0.009 | 0.009 |
| + | Gemini 2.5 Flash | 0.009 | 0.609 | 0.364 | 0.009 | 0.009 | 0.000 | 0.829 | 0.571 | 0.000 | 0.009 |
| Reviewer | Haiku 3.5 | 0.000 | 0.587 | 0.346 | 0.000 | 0.000 | 0.009 | 0.910 | 0.633 | 0.009 | 0.009 |
| | GPT-4o | 0.000 | 0.557 | 0.341 | 0.000 | 0.000 | 0.000 | 0.017 | 0.012 | 0.000 | 0.000 |
| | Gemma-2-9B-IT | 0.000 | 0.420 | 0.220 | 0.000 | 0.000 | 0.000 | 0.600 | 0.480 | 0.000 | 0.000 |

| Variation | Model | \multicolumn{5}{c}{**MetaOpenFOAM**} | | | | | \multicolumn{5}{c}{**Foam-Agent**} | | | | |
|---|---|---|---|---|---|---|---|---|---|---|---|
| | | $M_{\text{exec}}$ | $M_{\text{struct}}$ | $M_{\text{file}}$ | $M_{\text{NMSE}}$ | Success Ratio | $M_{\text{exec}}$ | $M_{\text{struct}}$ | $M_{\text{file}}$ | $M_{\text{NMSE}}$ | Success Ratio |
| | Sonnet 3.5 | 0.125 | 0.775 | 0.599 | 0.125 | 0.125 | 0.625 | 0.792 | 0.621 | 0.406 | 0.250 |
| RAG | o3-mini | 0.125 | 0.665 | 0.484 | 0.125 | 0.125 | 0.312 | 0.734 | 0.597 | 0.219 | 0.187 |
| + | Gemini 2.5 Flash | 0.000 | 0.796 | 0.586 | 0.000 | 0.000 | 0.062 | 0.692 | 0.511 | 0.000 | 0.000 |
| Reviewer | Haiku 3.5 | 0.062 | 0.646 | 0.498 | 0.031 | 0.000 | 0.188 | 0.840 | 0.609 | 0.188 | 0.187 |
| | GPT–4o | 0.000 | 0.514 | 0.430 | 0.000 | 0.000 | 0.375 | 0.805 | 0.634 | 0.312 | 0.250 |
| | Gemma-2-9B-IT | 0.000 | 0.540 | 0.410 | 0.000 | 0.000 | 0.000 | 0.630 | 0.450 | 0.000 | 0.000 |
| | Sonnet 3.5 | 0.000 | 0.743 | 0.594 | 0.000 | 0.000 | 0.188 | 0.771 | 0.609 | 0.156 | 0.125 |
| RAG | o3–mini | 0.000 | 0.746 | 0.535 | 0.000 | 0.000 | 0.000 | 0.702 | 0.566 | 0.000 | 0.000 |
| + | Gemini 2.5 Flash | 0.000 | 0.688 | 0.518 | 0.000 | 0.000 | 0.000 | 0.666 | 0.496 | 0.000 | 0.000 |
| No Reviewer | Haiku 3.5 | 0.000 | 0.654 | 0.518 | 0.000 | 0.000 | 0.000 | 0.801 | 0.583 | 0.000 | 0.000 |
| | GPT–4o | 0.000 | 0.733 | 0.603 | 0.000 | 0.000 | 0.000 | 0.744 | 0.594 | 0.000 | 0.000 |
| | Gemma-2-9B-IT | 0.000 | 0.530 | 0.400 | 0.000 | 0.000 | 0.000 | 0.610 | 0.440 | 0.000 | 0.000 |
| | Sonnet 3.5 | 0.375 | 0.655 | 0.451 | 0.344 | 0.187 | 0.250 | 0.806 | 0.592 | 0.188 | 0.125 |
| No RAG | o3–mini | 0.250 | 0.649 | 0.372 | 0.062 | 0.000 | 0.000 | 0.710 | 0.420 | 0.000 | 0.000 |
| + | Gemini 2.5 Flash | 0.000 | 0.685 | 0.407 | 0.000 | 0.000 | 0.000 | 0.702 | 0.410 | 0.000 | 0.000 |
| Reviewer | Haiku 3.5 | 0.000 | 0.718 | 0.456 | 0.000 | 0.000 | 0.000 | 0.825 | 0.511 | 0.000 | 0.000 |
| | GPT–4o | 0.188 | 0.658 | 0.415 | 0.000 | 0.000 | 0.000 | 0.710 | 0.443 | 0.000 | 0.000 |
| | Gemma-2-9B-IT | 0.000 | 0.500 | 0.360 | 0.000 | 0.000 | 0.000 | 0.590 | 0.400 | 0.000 | 0.000 |

### A.4.2 REASONS FOR FAILURE

We examine the common reasons for failure of execution for the cases in *FoamBench Basic* dataset for the two frameworks with the best performing model (Sonnet 3.5) in Figure 10. The mentioned reasons can be further elaborated as:

- Inconsistent Patch or Patch Field: This error means that certain boundaries that was defined in the boundary conditions file does not exist in the mesh file.

- File Not Found: This happens when a certain file, example: `blockMeshDict`, `controlDict` etc, required for the OpenFoam run is not found among the case files.

- Undefined keyword: This happens when certain keywords like the flux schemes or parameter values are not defined appropriately. The LLM will good knowledge about the OpenFoam to decided, which flux schemes are to be defined based on the solver that is being used and the variables that are being solved for.

- Numerical Instability: This occurs when the chosen numerical schemes, time step size, or boundary/initial conditions lead to unstable simulations, often causing divergence or NaN values in the solution. It typically arises from violating stability criteria (e.g., CFL condition) or poor discretization choices that amplify numerical errors during time integration.

- Geometry/Mesh Error: These errors stem from issues in mesh generation or geometry definition, such as non-orthogonal cells, skewed elements, or overlapping/missing faces. They can cause solver initialization to fail or lead to inaccurate or unphysical results during the simulation.

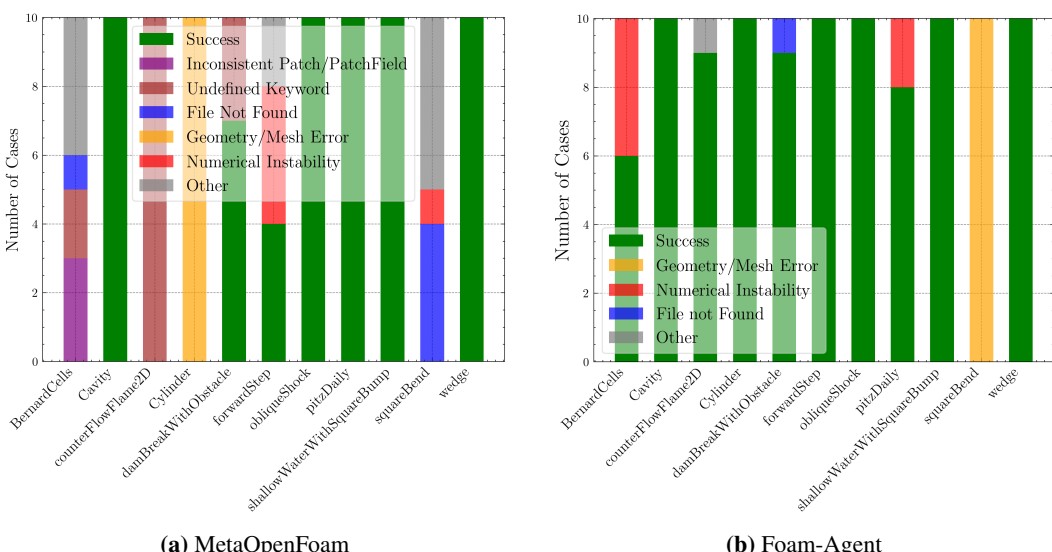

(a) MetaOpenFoam        (b) Foam-Agent

**Figure 10:** Common Reasons for execution failure found in MetaOpenFoam and Foam-Agent with RAG and Reviewer and using Sonnet 3.5 as the prompt model.

To qualitatively understand the impact of RAG and Reviewer in mitigating execution errors, we analyze the common failure reasons under three configurations: 1. Without RAG and Without Reviewer (zero-shot LLM), 2. With RAG and Without Reviewer and 3. Without RAG and With Reviewer, in Foam-Agent using Claude Sonnet 3.5 as the prompt model. The results are shown in fig. 11. We observe that RAG primarily mitigates configuration-related errors, such as missing physical properties, turbulence models, or undefined keywords, by providing accurate solver templates and reference parameters. In contrast, the Reviewer component reduces reasoning and consistency errors, such as mismatched boundary conditions or invalid inter-file dependencies. When combined, RAG offers factual grounding while the Reviewer enforces structural and physical consistency, together transforming failures from non-runnable setups into executable and physically valid simulations.

### A.4.3 TOKEN USAGE

The token usage statistics of the two frameworks in combination with the different models is shown in Table 8.

### A.4.4 PROMPT ENGINEERING

We investigated the role of prompting through a focused experiment on FOAMBENCH Advanced. Human-authored prompts were iteratively refined using the O3 reasoning model, and five validated

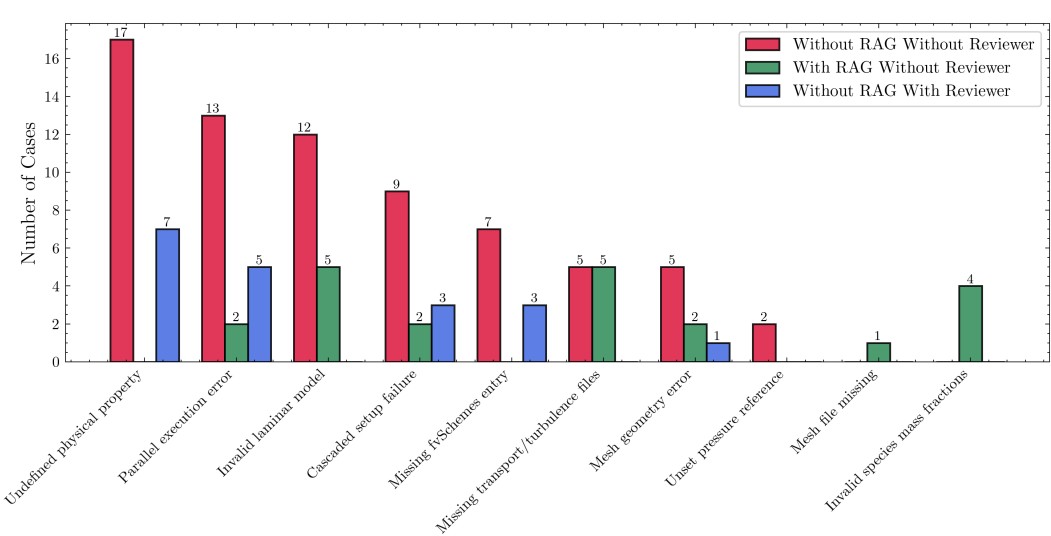

**Figure 11:** Common failure reasons found in Foam-Agent under three configurations 1. Without RAG and Without Reviewer (zero-shot LLM), 2. With RAG and Without Reviewer and 3. Without RAG and With Reviewer, using Claude Sonnet 3.5 as the prompt model.

**Table 8:** Average total token usage, API cost ($) as of May 2025, and loop counts per model variant for each agentic framework. Average is over all cases in *FoamBench*.

| Variation | Model | MetaOpenFOAM | | | | Foam-Agent | | | |
|---|---|---|---|---|---|---|---|---|---|
| | | Prompt | Completion | Cost ($) | Loop | Prompt | Completion | Cost ($) | Loop |
| RAG + Reviewer | Sonnet 3.5 | 63061.55 | 8337.20 | 1.19 | 7 | 378848.37 | 9346.10 | 6.56 | 2 |
| | o3-mini | 60273.86 | 45325.69 | 0.29 | 8 | 197334.74 | 6549.15 | 0.27 | 4 |
| | Gemini 2.5 Flash | 50603.35 | 8454.29 | 0.01 | 9 | 47929.00 | 4127.96 | 0.01 | 2 |
| | Haiku 3.5 | 29021.51 | 6499.74 | 0.06 | 9 | 426189.16 | 18235.88 | 0.43 | 5 |
| | GPT-4o | 73603.88 | 9109.8 | 0.28 | 9 | 147011.94 | 5702.80 | 0.42 | 9 |
| | Gemma-2-9B-IT | 35487.00 | 7322.00 | - | 10 | 331582.00 | 10322.00 | - | 10 |
| RAG + No Reviewer | Sonnet 3.5 | 10922.04 | 5526.07 | 0.13 | 1 | 278880.44 | 7358.98 | 5.0 | 1 |
| | o3-mini | 10128.16 | 24162.77 | 0.12 | 1 | 121070.15 | 3154.96 | 0.16 | 1 |
| | Gemini 2.5 Flash | 129137.86 | 5935.76 | 0.02 | 1 | 129137.86 | 5935.76 | 0.02 | 1 |
| | Haiku 3.5 | 11103.52 | 5563.83 | 0.06 | 1 | 110582.05 | 7818.34 | 0.50 | 1 |
| | GPT-4o | 9251.12 | 4662.87 | 0.07 | 1 | 80967.33 | 4389.14 | 0.24 | 1 |
| | Gemma-2-9B-IT | 8722.00 | 6670.00 | - | 1 | 98431.00 | 8888.00 | - | 1 |
| No RAG + Reviewer | Sonnet 3.5 | 39030.58 | 13369.42 | 0.54 | 9 | 92618.95 | 13369.42 | 1.28 | 4 |
| | o3-mini | 47062.19 | 47868.35 | 0.33 | 10 | 89131.14 | 5319.50 | 0.15 | 4 |
| | Gemini 2.5 Flash | 50253.88 | 10123.70 | 0.01 | 10 | 40675.74 | 6415.07 | 0.01 | 2 |
| | Haiku 3.5 | 24907.50 | 6663.95 | 0.04 | 10 | 111263.76 | 14704.62 | 0.61 | 4 |
| | GPT-4o | 52901.2 | 7958.66 | 0.21 | 10 | 202418 | 10972 | 0.55 | 4 |
| | Gemma-2-9B-IT | 29888.00 | 7799.00 | - | 10 | 95112.00 | 16444.00 | - | 10 |

variants were tested on Claude Sonnet 3.5 in a zero-shot setting (i.e., without retrieval augmentation and/or the Reviewer). The best variant improved success rate only marginally—from 0.007 to 0.012 Table 9, indicating that the original prompts were already effective and that extensive manual tuning provided limited additional benefit.

**Table 9:** FoamBench Advanced metrics for the best prompt variant (zero-shot, no RAG/Reviewer).

| Dataset | $M_{\text{exec}}$ | $M_{\text{struct}}$ | $M_{\text{file}}$ | $M_{\text{NMSE}}$ | Success Rate |
|---|---|---|---|---|---|
| FoamBench Advanced | 0.034 | 0.769 | 0.588 | 0.012 | 0.012 |

We have not conducted an extensive ablation study on prompt engineering. While prompt optimization is typically useful in large language model applications, our experience indicates that advanced methods such as retrieval-augmented generation (RAG) and the Reviewer tool have a substantially larger impact on downstream success rates. Indeed, these components increase Claude Sonnet 3.5's success on FOAMBENCH to roughly 0.25, far exceeding the modest gains achievable through manual prompt refinement. Nevertheless, to ensure fair comparison, *all models were evaluated under identical prompt settings*, so that differences in performance reflect model capability and auxiliary tooling rather than prompt variability.

## A.5 SOLUTION COMPARISON

### A.5.1 CFDQUERY

**Question:**
Which of the following is closest to the correct modified equation for the discretized 1D advection equation using second-order central difference in space and third-order Runge-Kutta (RK3) in time?

**Options:**

1. $\dfrac{\partial u}{\partial t} + a\dfrac{\partial u}{\partial x} = \dfrac{a\Delta x^2}{6}\dfrac{\partial^3 u}{\partial x^3} + \mathcal{O}(\Delta x^3)$

2. $\dfrac{\partial u}{\partial t} + a\dfrac{\partial u}{\partial x} = \dfrac{a\Delta x^2}{2}\dfrac{\partial^2 u}{\partial x^2} + \mathcal{O}(\Delta x^3)$

3. $\dfrac{\partial u}{\partial t} + a\dfrac{\partial u}{\partial x} = -\dfrac{a\Delta x^2}{6}\dfrac{\partial^3 u}{\partial x^3} + \dfrac{a\Delta t^2}{6}\dfrac{\partial^3 u}{\partial t^3} + \mathcal{O}(\Delta x^3)$

4. $\dfrac{\partial u}{\partial t} + a\dfrac{\partial u}{\partial x} = \dfrac{a\Delta x^2}{6}\dfrac{\partial^3 u}{\partial x^3} - \dfrac{a^3\Delta t^2}{6}\dfrac{\partial^3 u}{\partial x^3} + \mathcal{O}(\Delta x^3)$

**Correct Answer:** Option 4
**Model Responses:**

- **Sonnet 3.5:** Option 4 ✓
- **o3-mini:** Option 3 ✗
- **Gemini 2.5 Flash:** Option 4 ✓
- **Haiku 3.5:** Option 1 ✗
- **GPT-4o:** Option 3 ✗
- **Gemma-2-9B-IT:** Option 1 ✗

### A.5.2 CFDCODEBENCH

The visual comparison of the model produced results and the ground truth solution at the final timestep for the 1D Burgers equation is shown in Figure 12 and for the 2D convection equation is given in Figure 13. Models such as o3-mini, Haiku 3.5 and Gemini 2.5 Flash is able to closely match the ground truth solution for the 1D Burgers equation. Sonnet 3.5 is able to match the solution near the shock, but does not get the boundary conditions right.

In the solution for 2D convection, Sonnet 3.5 seems to have some numerical instability, while the other models seems to have a decent prediction of the x directional velocity.

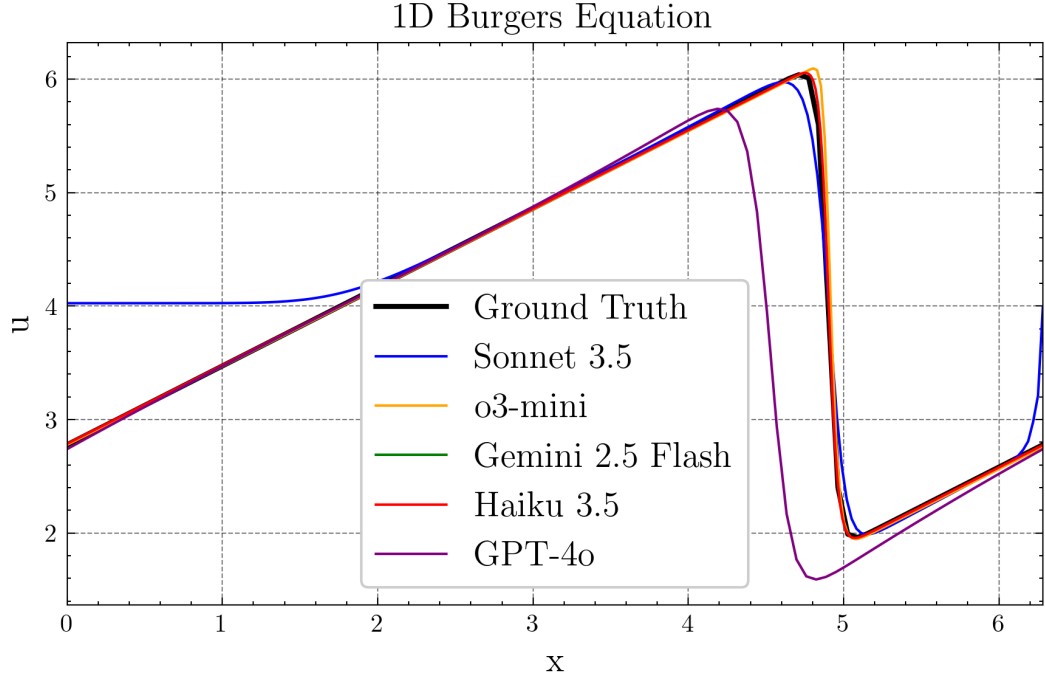

**Figure 12:** Solution comparison at the final time step for 1D Burgers equation

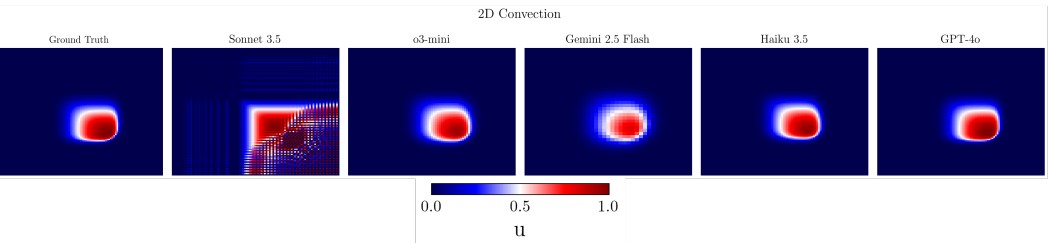

**Figure 13:** X direction velocity ($u$) comparison at the final time step for 2D Convection equation

### A.5.3 FOAMBENCH

Figure 14 and Figure 15 shows the comparison between the results from the two frameworks (*MetaOpenFOAM* and *Foam-Agent*) for the Cavity and forwardStep case respectively. The results from **Foam-Agent** is more similar to the ground truth in the Cavity case, while both the frameworks does well in the case of forwardStep.

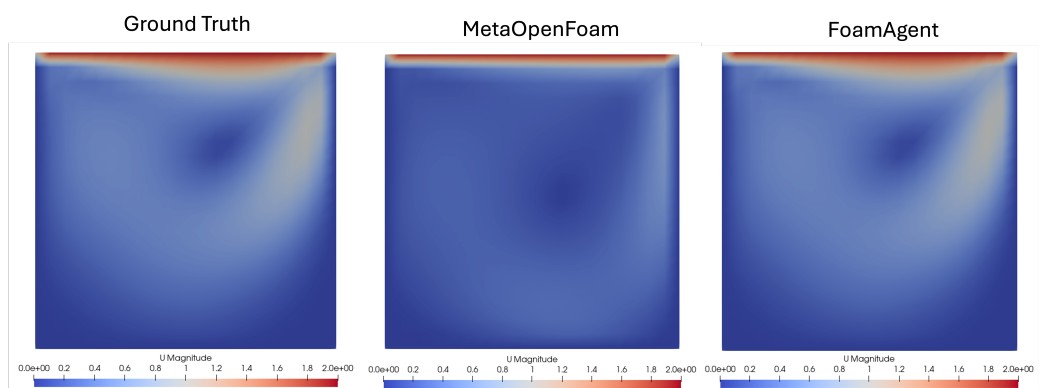

**Figure 14:** Comparison of velocity magnitude at the final timestep for 2D `Cavity` case.

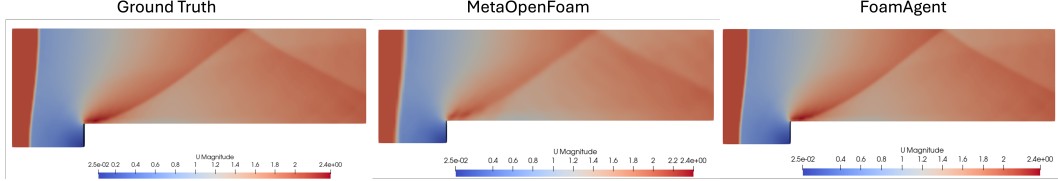

**Figure 15:** Comparison of velocity magnitude at the final timestep for 2D `forwardStep` case.

