# OpenReview forum: "CFDLLMBench: A Benchmark Suite for Evaluating Large Language Models in Computational Fluid Dynamics"
_ICLR.cc/2026/Conference — Submitted to ICLR 2026_

### Official Review · Reviewer_GN2B · 2025-10-20

**Soundness:** 3
**Presentation:** 3
**Contribution:** 3
**Rating:** 6
**Confidence:** 3

**Summary:**

This paper introduces CFDLLMBench, a benchmark suite designed to holistically evaluate the comprehensive capabilities of LLMs in the specialized domain of CFD. The authors posit that while LLMs excel at general nlp tasks, their potential for automating complex scientific computing workflows remains insufficiently explored. CFDLLMBench consists of three components with progressively increasing difficulty: CFDQuery, CFDCodeBench, and FoamBench. Through rigorous experimental design and in-depth case studies, the paper reveals key shortcomings of current LLMs when applied to the scientific computing domain and provides directions for future research.

**Strengths:**

1.  This paper is the first to propose a comprehensive, layered LLM benchmark specifically for the CFD domain. CFD, a field that demands a high degree of physical understanding, numerical methods knowledge, and software operation skills, serves as an excellent scenario for testing the scientific capabilities of LLMs. This work fills a gap in existing LLM benchmarks concerning the automation of complex scientific workflows.

2.  The knowledge-reasoning-practice structure of CFDLLMBench is exceptionally well-designed. This progression from simple to complex allows for a clear dissection of the LLM's capability boundaries, revealing precisely at which stage the model encounters a bottleneck.

3.  The paper introduces evaluation metrics closely tied to real-world CFD practices: code executability, physical result accuracy, and numerical convergence. This set of metrics significantly enhances the scientific rigor and credibility of the evaluation.

**Weaknesses:**

No obvious weaknesses were found; please refer to the Questions section. :-)

**Questions:**

1.  The definition of success rate seems very strict, requiring the three metrics $M_{exec}$, $M_{NMSE}$, and $M_{conv}$ to simultaneously achieve a perfect score. Have you considered an evaluation system for "partial success"? Such an analysis might offer a more fine-grained perspective for understanding the model's capabilities across different sub-tasks.


2.  The experimental results demonstrate that both RAG and the Reviewer are crucial. From a qualitative perspective, what are the primary problems that each of these components addresses? For instance, when RAG is absent, is the model's most common failure the selection of the wrong solver or the inability to correctly set parameters?

---

> ### Author Response · Authors · 2025-11-19
> **Response to the Reviewer Comments and Questions**
>
> **We thank the reviewer for the thoughtful and encouraging assessment and for giving our paper a good score. We are glad that you found the structure, rigor, and contribution of CFDLLMBench to be valuable.**
>
> **Minor**
>
> > **Q1.** **The definition of success rate seems very strict, requiring the three metrics to simultaneously achieve a perfect score. Have you considered an evaluation system for "partial success"? Such an analysis might offer a more fine-grained perspective for understanding the model's capabilities across different sub-tasks.**
>
> **A1.** We agree that the current definition, requiring (1) executability, (2) physical correctness, and (3) numerical convergence to all succeed, is strict. However, each individual metric already reflects a form of partial success: executability captures syntactic and runtime validity, physical correctness measures the accuracy of the solution, and numerical convergence measures algorithmic fidelity. Our reported “Success Rate” simply represents the joint probability of all metrics passing. Hence, partial success can be inferred directly from the individual scores for executability, physical correctness, and convergence. These partial metrics are visualized in Figure 4 for CFDCodeBench and Tables 1 and 2 for FoamBench in the revised manuscript, with detailed scoring across different models in appendix section A.4 for FoamBench.
>
> > **Q2.** **The experimental results demonstrate that both RAG and the Reviewer are crucial. From a qualitative perspective, what are the primary problems that each of these components addresses? For instance, when RAG is absent, is the model's most common failure the selection of the wrong solver or the inability to correctly set parameters?**
>
> **A2.** Our post hoc analysis compares the most frequent failure causes across three configurations using Foam-Agent (Claude Sonnet 3.5): (1) without RAG and Reviewer, (2) with RAG only, and (3) with Reviewer only. We observe that **RAG** primarily mitigates configuration-related errors (e.g., missing physical properties, turbulence models, or undefined keywords) by providing accurate solver templates and reference parameters. In contrast, the **Reviewer** component reduces reasoning and consistency errors, such as mismatched boundary conditions or invalid inter-file dependencies. When combined, RAG offers factual grounding while the Reviewer enforces structural and physical coherence, together transforming many failures from non-runnable setups into executable and physically valid simulations. Further details on the specific errors can be found in Appendix A.4.2, Figure 11 of the revised manuscript.

---

> > ### Author Response · Authors · 2025-11-26
> >
> > Thank you again for your careful evaluation of our work. We understand that this is a busy period in the review process. If there are any additional questions, concerns, or points that would benefit from further clarification before the final decision, we would be glad to address them promptly.
> > Please feel free to let us know if any further information from our side would be helpful.

---

### Official Review · Reviewer_nPTb · 2025-10-27

**Soundness:** 3
**Presentation:** 3
**Contribution:** 3
**Rating:** 6
**Confidence:** 3

**Summary:**

The paper presents CFDLLMBench, a benchmark datset designed to evaluate large language models (LLMs) on tasks in computational fluid dynamics (CFD). This includes 1) CFDQuer : 90 multiple-choice questions testing graduate-level CFD knowledge, including fluid mechanics, numerical methods, and linear algebra 2) CFDCodeBench: 24 programming tasks requiring LLMs to generate Python code for solving 1D and 2D PDEs with specified boundary and initial conditions and 3) FoamBench: 126 OpenFOAM cases (110 basic, 16 advanced) requiring LLMs to generate complete input files and configurations for realistic CFD scenarios.

**Strengths:**

* The benchmark tasks are accopanied by tailored metrics for CFD (such as executability, numerical accuracy, convergence, and physical correctness).

* The paper covers diverse tasks, namely, conceptual understanding (CFDQuery), code generation (CFDCodeBench), and workflow automation (FoamBench).

* The works provides a benchmark for OpenFOAM, a widely adopted CFD suite for simulation tasks.

**Weaknesses:**

* Dataset statistics and distributions of problem types are unclear. It would be helpful for understanding the benchmark to provide sub-categories for problem difficulty (such as based on PDE characteristics, initial conditions, etc.)

* L253 - 257: The quality assessment of the dataset is unclear. How many human hours went into quality assessment? What criteria were used?

**Questions:**

* How does this benchmark account for different (but valid) numerical algorithms for a PDE problem which may have better or worse performance depending on the PDE and evaluation score in question ?

* If the dataset was constructed from web-scraped content, how to decouple "the LLMs to have deep knowledge about topics in CFD like linear algebra, numerical methods and fluid dynamics (L 188)" with possible test-set leakage with LLM training?

---

> ### Author Response · Authors · 2025-11-19
> **Response to the Reviewer Comments and Questions**
>
> **We thank the reviewer for the constructive feedback, for recognizing the diversity and relevance of CFDLLMBench, and for giving our paper a good score.**
>
> **Major**
>
> > **Q1.** **Dataset statistics and distributions of problem types are unclear. It would be helpful for understanding the benchmark to provide sub-categories for problem difficulty (such as based on PDE characteristics, initial conditions, etc.)**
>
> **A1.**  We agree that understanding the distribution of problem types is important. The detailed breakdowns for **CFDCodeBench** and **FoamBench** are provided in Appendix A.3; here we summarize the key statistics:
>
> - **CFDCodeBench:** 24 PDE problems, including
>   - 1D vs 2D: **8 (33%) 1D**, **16 (67%) 2D**
>   - Linear vs nonlinear vs turbulent/complex: **9 linear (38%)**, **11 nonlinear (46%)**, **4 turbulent/complex (16%)**
>   - Steady vs unsteady: **7 steady**, **17 unsteady**
>   These cover 1D/2D convection–diffusion, Burgers, Euler, Poisson/Laplace, stability/eigenvalue problems, and laminar/turbulent flow setups.
>
> - **FoamBench Basic:** **11 base OpenFOAM tutorials** (e.g., cavity, cylinder, pitzDaily, damBreakWithObstacle, wedge, obliqueShock, shallowWaterWithSquareBump), each varied across **10 parameter sets**, yielding **110 cases**.
>   The set spans **laminar, turbulent, compressible, multiphase, and reacting flows**, and **internal, external, and free-surface geometries**, providing diverse difficulty in terms of physics, boundary conditions, and numerical stability.
>
> We have also added the overall distribution of the cases in CFDLLMBench to the revised manuscript as Figure 3.
>
> > **Q2.** **The quality assessment of the dataset is unclear. How many human hours went into quality assessment? What criteria were used?**
>
> **A2.**  We used expert review with approximate time budgets per item:
>
> - **CFDQuery:** Trivially solvable or low-difficulty questions were removed. Each remaining question underwent ~**30 minutes** of expert review to verify solution correctness and remove ambiguity in both questions and answers.
> - **CFDCodeBench:** Curated prompts and reference codes were reviewed for inconsistencies or unclear instructions, taking ~**10 minutes per case**.
> - **FoamBench:** OpenFOAM prompts and simulation setups were checked for clarity, reproducibility, and numerical stability. Ground-truth tutorial cases were re-executed and validated against their prompts, requiring ~**one hour per case type** (including checks over its variations).
>
> **Minor**
>
> > **Q3.** **How does this benchmark account for different (but valid) numerical algorithms for a PDE problem which may have better or worse performance depending on the PDE and evaluation score in question?**
>
> **A3.**  Our evaluation focuses on the **results**, not the specific numerical algorithm. If an LLM uses a different but valid scheme (e.g., finite difference vs finite volume, different time integrators), the solution is accepted as long as:
> - the numerical error remains below the specified tolerance, and
> - the solution converges without instability.
> Thus, we do **not** penalize algorithmic diversity; we only evaluate correctness and stability of the outputs. We emphasize the same in the revised manuscript (L323-325).
>
> > **Q4.** **If the dataset was constructed from web-scraped content, how to decouple "the LLMs to have deep knowledge about topics in CFD like linear algebra, numerical methods and fluid dynamics (L 188)" with possible test-set leakage with LLM training?**
>
> **A4.**  To mitigate test-set leakage and overlap with web-scraped content, we applied two safeguards:
>
> - We **removed any question that was trivially answered by all major LLMs** (GPT-4, Claude, Gemini, etc.) during pilot runs.
> - We **rephrased, modified, or combined** existing concepts so that questions emphasize reasoning and numerical understanding rather than memorized text patterns.
>
> While full elimination of overlap with pretraining data is impossible, these steps ensure CFDLLMBench primarily measures **reasoning and generalization**, not simple recall.

---

> > ### Author Response · Authors · 2025-11-26
> >
> > Thank you again for your careful evaluation of our work. We understand that this is a busy period in the review process. If there are any additional questions, concerns, or points that would benefit from further clarification before the final decision, we would be glad to address them promptly.
> > Please feel free to let us know if any further information from our side would be helpful.

---

### Official Review · Reviewer_4Pxf · 2025-10-31

**Soundness:** 2
**Presentation:** 2
**Contribution:** 1
**Rating:** 2
**Confidence:** 5

**Summary:**

The authors proposed a set of dataset and benchmark problems for evaluating different large language models (LLMs) such as ChatGPT, Gemini, Sonnet, etc, for evaluating computational fluid dynamics (CFD) problems. They proposed matrices for measuring the success rate of these LLMs and made a comparison through visual graphs and numerical values.

**Strengths:**

1) The topic is relevant to LLMs, which nowadays is hot.
2) Nice plots and figures
3) Good amount of supportive materials
4) Good writing

**Weaknesses:**

The current manuscript is a technical report and does not present any novel algorithm or methodology. Even in the context of a technical report (evaluating the performance of different LLMs for CFD), this report claims inaccurate information. For example, the authors wrote in conclusion (and similarly in introduction) that:

"In this work, we introduced CFDLLMBench, the first benchmark to holistically evaluate graduate level knowledge, numerical and physical reasoning, and practical simulation capabilities of LLMs for CFD"

It is not true and they are not the first. I believe that the manuscript suffers from the adequate and appropriate literature reviews. Pioneers of people who evaluated LLMs for CFD have been missed from the reference section and the authors claimed that they are first scientists performing such experiments. For example, please see the following journal paper:

https://www.dl.begellhouse.com/journals/558048804a15188a,498820861ef102d2,1255e053242c9a40.html

“ChatGPT for programming numerical methods”

It was just published in 2023, a few months after releasing ChatGPT by OpenAI.

In pages 23 and 27 of this journal paper, the authors evaluated the performance of ChatGPT for coding the 1D compressible flow and in pages 57, 58, and 59, they investigated the performance of ChatGPT for coding 2D incompressible flow in C++. These are two examples for CFD.

There are other journal papers that have been missing from this manuscript, for example see:

https://www.sciencedirect.com/science/article/pii/S2095034925000157

“DeepSeek vs. ChatGPT vs. Claude: A comparative study for scientific computing and scientific machine learning tasks”

A few more points to be considered:

--> The quality of Figure 1 is low, and some text cannot be read.

--> In Figure 2, we observe that the authors listed the performance of o3-mini, which has been removed by Open AI for a few months and instead we have GPT 5, which has been missed from the current manuscript. Of course, I understand that this is because the fast movement of AI technology, but this fact supports me total judgment about the manuscript, that this is simply a technical report and not a scientific paper suitable for publication in ICLR 2026. Perhaps, it could be considered for the workshop sections.

**Questions:**

The state of art of using LLM for CFD is to develop domain-knowledge LLM for CFD. I suggest considering this direction. Evaluating the current LLMs and providing benchmarks is good, but it is not really the concern of CFD community. My main questions and concerns have been listed in the previous box. Thanks for your submission to ICLR.

---

> ### Author Response · Authors · 2025-11-19
> **Response to the Reviewer Comments and Questions**
>
> **We thank the reviewer for their feedback and appreciate the opportunity to clarify the novelty and purpose of our work.**
> **Major**
>
> > **Q1.** **The current manuscript is a technical report and does not present any novel algorithm or methodology.**
>
> **A1.**  Our submission introduces **CFDLLMBench**, a benchmarking suite designed to evaluate large language models (LLMs) on domain-specific scientific reasoning and numerical simulation tasks in CFD. While we do not propose a new algorithm, we contribute:
> - **High-quality datasets** (CFDQuery, CFDCodeBench, FoamBench)
> - **Unified evaluation metrics and scoring matrices**
> - **Standardized workflows and SOTA model comparisons**
>
> Thus, our work is **not a mere technical report**, but benchmark infrastructure that enables AI researchers to build, evaluate, and compare CFD-focused LLM systems. Without such benchmarks, researchers would rely on inconsistent datasets, evaluation setups, and scoring standards, making cross-study comparison difficult.
>
> Benchmark creation of this kind is explicitly within **ICLR’s Datasets & Benchmarks subject area**. Accepted ICLR works such as **Open-CK** [1], **UGMathBench**[2], and **BEND**[3] focus on datasets and metrics rather than new algorithms; our work follows this precedent in the CFD domain.
>
> > **Q2.** **"In this work, we introduced CFDLLMBench, the first benchmark to holistically evaluate graduate level knowledge, numerical and physical reasoning, and practical simulation capabilities of LLMs for CFD". It is not true and they are not the first.**
>
> **A2.**  The reviewer points out two related works [4][5]. However, these are **case-based demonstrations**, not benchmark suites. They do not provide standardized datasets or code templates, unified quantitative metrics, or reproducible evaluation pipelines.
>
> By contrast, **CFDLLMBench** provides public benchmark datasets, a unified scoring matrix for physical reasoning, numerical accuracy, and simulation success, and evaluates multiple commercial and open LLMs under identical protocols with automated grading tools. The following table summarizes the distinctions:
>
> | Feature                        | *ChatGPT for Programming Numerical Methods* | *DeepSeek vs ChatGPT vs Claude* | *CFDLLMBench (ours)* |
> | ------------------------------ | ------------------------------------------- | -------------------------------- | -------------------- |
> | Theoretical CFD questions      | No                                          | No                               | **Yes** (90+ CFD theory Qs) |
> | PDEs considered                | 2D Poisson & diffusion; 2D incompressible NS; 1D compressible Euler | 2D Poisson + a few SciML/PDE demos | 20+ 1D/2D CFD PDE tasks (incompressible, compressible, multiphase, reactive) |
> | Tool-use / agentic evaluation  | No, single-model code gen/debug only       | No, code/training only          | **Yes**, 110+ OpenFOAM cases for agentic tool use (Foam-Agent + FoamBench) |
> | Benchmark suite + metrics      | No, case study, no standardized splits or metrics | No, illustrative tasks, not a packaged benchmark | **Yes**, defined tasks, splits, metrics, and automated evaluation |
> | Open dataset / code            | **Yes** (code only, not structured as benchmark) | No public benchmark dataset/code | **Yes**, datasets + evaluation code + Dockerized pipeline |
>
> Hence we re-emphasize that **CFDLLMBench** is the first holistic and reproducible benchmark suite that spans theoretical, numerical, and practical simulation capabilities of LLMs for CFD. We have also added the suggested papers to our literature review.
>
> **Minor**
>
> > **Q1.** **Quality of figure 1**
>
> **A1.**  We have replaced Figure 1 with a higher-resolution version and added illustrative icons to better convey the intended meaning in the revised pdf.
>
> > **Q2.** **In Figure 2, we observe that the authors listed the performance of o3-mini, which has been removed by Open AI for a few months and instead we have GPT 5, which has been missed from the current manuscript.**
>
> **A2.**  We evaluated a range of state-of-the-art closed- and open-source models available at the time of experimentation. Newer models such as GPT-5 and Claude Sonnet 4.0 require substantially more time and monetary resources to evaluate. We are currently running experiments with these models and will incorporate the results as soon as they are available.
>
> *We agree with the reviewer that developing domain-knowledge LLMs for CFD is an important future direction. The proposed benchmark directly enables this line of research by providing a standardized suite to evaluate and improve such models.*
>
> [1] *https://openreview.net/forum?id=A23C57icJt*
>
> [2] *https://openreview.net/forum?id=fovPyqPcKY*
>
> [3] *https://openreview.net/forum?id=uKB4cFNQFg*
>
> [4] *ChatGPT for Programming Numerical Methods* (Begell House, 2023).
>
> [5] *DeepSeek vs. ChatGPT vs. Claude: A Comparative Study for Scientific Computing and Scientific ML Tasks* (ScienceDirect, 2025).

---

> > ### Author Response · Authors · 2025-11-26
> >
> > Thank you again for your careful evaluation of our work. We understand that this is a busy period in the review process. If there are any additional questions, concerns, or points that would benefit from further clarification before the final decision, we would be glad to address them promptly.
> > Please feel free to let us know if any further information from our side would be helpful.

---

### Author Response · Authors · 2025-12-02
**Rebuttal Summary (1/2)**

Dear Area Chair,

Thank you for taking time to review our submission under these special circumstances. This note summarizes the main discussion points raised in the reviews and our corresponding clarifications.

**1. Track fit and nature of the contribution**
- **Concern:** The paper is “just a technical report” without a new algorithm and therefore not suitable for ICLR.
- **Our clarification:** The submission is explicitly to the *Datasets & Benchmarks* subject area. The main contribution is benchmark infrastructure:
  - Three curated components: **CFDQuery**, **CFDCodeBench**, and **FoamBench**.
  - Unified evaluation metrics: executability, numerical accuracy, numerical convergence, and physical correctness.
  - Reproducible workflows with open-source code and Dockerized pipelines for multiple LLMs.
  This is aligned with prior ICLR D&B papers whose primary novelty is standardized datasets, metrics, and baselines rather than new algorithms.

**2. Novelty vs prior LLM for CFD works**
- **Concern:** We are “not the first” to evaluate LLMs for CFD; prior papers already exist.
- **Our clarification:** The cited works (e.g., *ChatGPT for Programming Numerical Methods*; *DeepSeek vs ChatGPT vs Claude for scientific computing*) are **case studies**, not benchmark suites. They:
  - Consider a small number of PDE/CFD tasks and one or a few models.
  - Do not define standardized splits, unified scoring criteria, or automated evaluation pipelines.
  - Do not evaluate agentic tool use or end-to-end OpenFOAM workflows.
  In contrast, **CFDLLMBench**:
  - Covers graduate-level CFD theory (90+ questions), 24 diverse 1D/2D PDE coding tasks, and 110+ OpenFOAM cases.
  - Releases datasets, evaluation code, and a Dockerized pipeline.
  We have added these works to the related work section and clarified our claim as being the first **holistic, reproducible benchmark suite** spanning CFD knowledge, numerical reasoning, and practical simulation workflows.

**3. Dataset coverage, statistics, and quality control**
- **Concern:** Distributions of problem types and the dataset quality assessment process were unclear.
- **Our clarification (now reflected in the revision):**
  - **CFDCodeBench (24 PDE tasks):** 8×1D vs 16×2D; 9 linear, 11 nonlinear, 4 turbulent/complex; 7 steady vs 17 unsteady, including convection–diffusion, Burgers, Euler, Poisson/Laplace, eigenvalue/stability, laminar/turbulent setups.
  - **FoamBench Basic (110 cases):** Built from 11 canonical OpenFOAM tutorials (cavity, cylinder, pitzDaily, damBreakWithObstacle, wedge, shallow water, shocks, etc.) with 10 parameter variations each, spanning laminar/turbulent, compressible/incompressible, multiphase/reacting, and internal/external/free-surface flows. A summary figure is added in the main text.
  - **Quality control:**
    - CFDQuery: Each non-trivial question underwent ~30 minutes of expert review to verify correctness and remove ambiguity.
    - CFDCodeBench: Prompts and reference codes received ~10 minutes of expert review per task for clarity and consistency.
    - FoamBench: Each tutorial family (with its variations) was re-run and validated (~1 hour per case type) for clarity, reproducibility, and numerical stability.

---

### Author Response · Authors · 2025-12-02
**Rebuttal Summary (2/2)**

**4. Handling multiple valid algorithms and potential test-set leakage**
- **Concern 1:** How do we treat different but valid numerical schemes on the same PDE?
  - **Clarification:** We evaluate **outputs, not specific algorithms**. Any scheme (FD, FV, different time integrators, etc.) is accepted as long as the solution error is below a specified tolerance and the run converges. We explicitly emphasize this to avoid penalizing algorithmic diversity.
- **Concern 2:** Risk that web-scraped content overlaps with LLM pretraining data.
  - **Clarification:** Complete elimination of overlap is impossible, but we minimize it by:
    - Removing questions that all major LLMs answer trivially in pilot runs (likely memorized or too easy).
    - Rephrasing/combining sourced items so that questions require reasoning and numerical understanding rather than verbatim recall.
  Thus CFDLLMBench primarily measures reasoning and generalization rather than rote memorization.

**5. Strict success metric, partial success, and analysis of RAG/Reviewer**
- **Concern 1:** Success definition is very strict (all executability, physical correctness, and convergence must pass), with no explicit “partial success.”
  - **Clarification:** The strict success rate is intended as a high-bar, end-to-end metric. However, we already report the **individual metrics separately** (executability, physical correctness, convergence), which provide a fine-grained view of partial success. These per-metric results are visualized in Figure 4 (CFDCodeBench), Tables 1–2 (FoamBench), and Appendix A.4.
- **Concern 2:** What distinct roles do RAG and the Reviewer play in Foam-Agent?
  - **Clarification:** Our post hoc analysis for Claude Sonnet 3.5 shows:
    - **RAG** mainly fixes configuration errors (missing physical properties, turbulence models, undefined fields/keywords) by providing accurate templates and parameters.
    - **Reviewer** mainly fixes reasoning/consistency errors (mismatched boundary conditions, inconsistent parameters across files, invalid inter-file dependencies).
    When combined, they turn many non-runnable or physically invalid setups into executable, coherent simulations. We document this error breakdown in Appendix A.4.2 (Figure 11).

**6. Minor issues (figures and model set)**
- We have improved Figure 1 with a higher-resolution, more legible version.
- We clarify that the evaluated models correspond to those available at experiment time (including o3-mini). Newer models (e.g., GPT-5, newer Claude variants) are more expensive to evaluate; we are running additional experiments and will incorporate them as resources permit.

Overall, our rebuttal clarifies that CFDLLMBench is a rigorously constructed, reproducible **benchmark suite** and not just a case study, intended as shared infrastructure for building and evaluating domain-specialized CFD LLMs.

---

### Meta-Review · Area_Chair_FJRk · 2026-01-03

**Summary:**

This proposed benchmark suite evaluates LLMs on CFD knowledge, PDE code generation, and OpenFOAM workflows with executability/accuracy/convergence metrics.

**Reviewer Concerns:**

Main concerns from reviewers include:

1. Reviewers question novelty/fit (too “technical report”) and dispute the “first holistic benchmark” claim (missing related work).

2. Reviewers request clearer dataset statistics and QA procedure/effort, and flag validity risks: multiple correct schemes, potential web-scrape leakage, and strict success metric definition/partial-success analysis.

**Reviewer Scores:**

Reviewer / Score

4Pxf	2

nPTb	6

GN2B	6

Average	4.666666667

No reviewers indicated to increase or decrease their scores.

---

### Decision · Program_Chairs · 2026-01-26

Reject